# Structure of the recombinant RNA polymerase from African Swine Fever Virus

Simona Pilotto ®[1,2], Michal Sýkora ®[1,2], Gwenny Cackett ®[1,2], Christopher Dulson ®[1] & Finn Werner ®[1] ✉

African Swine Fever Virus is a Nucleo-Cytoplasmic Large DNA Virus that causes an incurable haemorrhagic fever in pigs with a high impact on global food security. ASFV replicates in the cytoplasm of the infected cell and encodes its own transcription machinery that is independent of cellular factors, however, not much is known about how this system works at a molecular level. Here, we present methods to produce recombinant ASFV RNA polymerase, functional assays to screen for inhibitors, and high-resolution cryo-electron microscopy structures of the ASFV RNAP in different conformational states. The ASFV RNAP bears a striking resemblance to RNAPII with bona fide homologues of nine of its twelve subunits. Key differences include the fusion of the ASFV assembly platform subunits RPB3 and RPB11, and an unusual C-terminal domain of the stalk subunit vRPB7 that is related to the eukaryotic mRNA cap 2´-O-methyltransferase 1. Despite the high degree of structural conservation with cellular RNA polymerases, the ASFV RNAP is resistant to the inhibitors rifampicin and alpha-amanitin. The cryo-EM structures and fully recombinant RNAP system together provide an important tool for the design, development, and screening of antiviral drugs in a low biosafety containment environment.

RNA polymerases (RNAPs) of the double-psi beta-barrel (DPBB) family are evolutionary conserved in all domains of life and transcribe the cellular genomes of eukaryotes, archaea and bacteria[1]. In addition, some bacteriophages[2], viruses[3] and virus-like elements in yeast[4] encode distant relatives of cellular RNAPs, but their structure and functional characteristics are understudied compared to their cellular systems. This includes the molecular mechanisms of viral RNAP and the transcription cycle (initiation, elongation, and termination), promoter recognition, interactions with transcription factors, as well as the role of viral chromatin in the regulation of gene expression. African Swine Fever (ASF) is a haemorrhagic disease in domesticated and wild pigs with near 100 % fatality[5]. The causative agent of ASF is a double-stranded DNA (dsDNA) virus that belongs to the phylum of Nucleo-Cytoplasmic Large DNA Viruses (NCLDV). To date, the only structurally characterised NCLDV RNAP is from Vaccinia Virus (VACV)[6,7], a member of the *Poxviridae* family, which also includes Variola virus (Smallpox) and Monkeypox virus[8]. African Swine Fever Virus belongs to the

*Asfarviridae* family, it infects macrophages and monocytes of swine and related species such as wild boars, warthogs and African bushpigs[9]. In its native Sub-Saharan Africa, ASFV exhibits a sylvatic replication cycle and uses *Ornithodoros* soft ticks as vector[10]. The diversity of ASFV strains has arisen from gene duplications and deletions primarily at the flanks of its linear DNA genome as the virus has spread from Africa across Europe and Asia[11]. Unlike many other DNA viruses, most NCLDVs replicate in the cytoplasm, hence they have no access to the host's RNAP in the nucleus and encode a complete viral transcription machinery[12]. ASFV produces/expresses all enzymes required to synthesise 5´-capped and 3´-polyadenylated mRNAs that are translated by host ribosomes. This includes RNAP and initiation factors that enable a temporally regulated transcription programme over the infection time course, an mRNA capping enzyme (CE), a poly-adenylation polymerase, and a histone-like protein[12,13]. All factors that are required for transcription during the early stages of infection, minutes after virus particle ingress, are packaged into ASFV particles,

[1]Institute for Structural and Molecular Biology, Division of Biosciences, University College London, Gower Street, London WC1E 6BT, United Kingdom. [2]These authors contributed equally: Simona Pilotto, Michal Sýkora, Gwenny Cackett. ✉e-mail: f.werner@ucl.ac.uk

making early gene transcription independent of host factors. This is evidenced by fully transcription competent extracts prepared from virus particles[14].

VACV has the only structurally characterised NCLDV transcription system and serves as a useful starting point for the exploration of the unknown ASFV RNAP system. VACV core RNAP includes eight subunits (Rpo147, Rpo132, Rpo35, Rpo22, Rpo19, Rpo18, Rpo7, and Rpo30) with varying degree of homology to RNAPII (RPB1, 2, 3-11 fusion, 5, 6, 7, 10, and TFIIS, respectively) (Table 1). It forms multiple higher-order complexes with RNAP-associated proteins Rap94 (RNAP associated protein 94), E11, the CE subunits D1 and D12, NPH-I (nucleoside tri-phosphate phosphohydrolase 1), the VETF (VACV early transcription factor) initiation factors, and a [Gln]tRNA molecule. Cryo-EM structures of the VACV early preinitiation complexes have provided the first glimpses of the structural basis of NCLDV RNAP and transcription initiation[6,7,15]. However, all structural analyses of VACV RNAP are currently reliant on affinity-purified RNAP complexes isolated from HeLaS3 cells infected with VACV, which yield a diverse range of RNAP complexes that represent different stages of transcription complex assembly.

Due to the virulence of ASFV, all work with live virus must be carried out in high biosafety containment facilities, which is cumbersome and resource intensive. To overcome this limitation and to ensure full control over the complex composition of the ASFV RNAP, we adopted a fundamentally different strategy by co-expressing the RNAP subunits and factors in insect cells, aiming to create RNAP assemblies of increasing complexity in a step-by-step manner. Based on sequence predictions and confirmed by mass spectrometry of viral particles, the ASFV RNAP is comprised of eight RNAPII subunit homologues including RPB1, 2, a RPB3-11 gene fusion, 5, 6, 7, 9 and 10[13,16]. ASFV furthermore encodes factors with limited sequence similarity to structurally characterised VACV CE D1, NPH-I, and VETF initiation factors, but none with clear similarity to Rap94, D12, and E11. In research leading up to this study, we characterised the ASFV tranmscriptome, the temporal regulation of viral gene expression during early (5 hours post infection, hpi) and late infection (16 hpi), and the sequence motifs associated with ASFV promoters (early and late promoters) and transcription terminators[13,17,18].

Here, we report the successful production of recombinant catalytically active ASFV core RNAP in insect cells at yields suitable for functional characterisation and inhibitor screening without the need for containment facilities. Our results show that ASFV RNAP is resistant to RNAPII and bacterial RNAP inhibitors alpha-amanitin and rifampicin, respectively, in line with previous findings using ASFV RNA polymerase isolated from viral particles[19]. We have solved the cryo-EM structures of the RNAP with the clamp in closed and open conformations at 2.7 and 2.9 Å resolution, respectively. These structures reveal a strong overall conservation of all critical functional elements that characterise cellular DPBB RNAPs, including a common NCDLV fusion of subunits 3 and 11[3,6]. In addition, we found ASFV-specific structural features like altered rim helices and the RPB2 horn motif. A striking difference to cellular RNAPs is the fusion of the RPB7 subunit with a domain evolutionary related to the 2′-O-methyltransferase 1 (2′O-MTase 1) domain of the eukaryotic mRNA capping enzyme. These ASFV-specific adaptions may enable efficient RNAP assembly and biogenesis, as well as co-transcriptional mRNA capping and translation, respectively, thus providing selective advantages for viral gene expression.

## Results

### In vivo RNAP subunit co-expression and assembly

For clarity, we have adopted the yeast RNAPII nomenclature for the eight core ASFV subunits from the BA71V strain, preceded by the letter v for viral (Table 1). The assembly of multisubunit RNAPs is evolutionary conserved in bacteria, archaea, and eukaryotes[20,21]. RNAP biogenesis is nucleated by the formation of the assembly platform, the heterodimerisation of RPB3 and 11, followed by the incorporation of RPB10 and 12. In ASFV, this step corresponds to the heterodimerisation of the fused vRPB3-11 and 10. Subsequently, RPB2, is incorporated into the assembly platform, and finally RPB1 is added to form the RNAP in its minimal subunit configuration required for catalysis. As last step, the small auxiliary subunits RPB4, 5, 6, 7 and 9 are incorporated into the RNAP, where RPB6 acts as a chaperone for RPB1. Based on the RNAP assembly pathway and the inter-subunit interactions predicted from archaeo-eukaryotic RNAPs (vRPB2-9, vRPB1-5, and vRPB1-7), we designed two multicistronic recombinant baculovirus genomes (bacmids). The first bacmid encodes vRPB2, 3-11, 9 and 10, and a separate second bacmid vRPB1, 5, 6, and 7 (Fig. 1a). Both viruses were co-infected into insect cells, and the RNAP assemblies were isolated using a two-step purification protocol. After an initial affinity purification using a TEV-cleavable ZZ-tag[22] on the N-terminus of vRPB2, the TEV cleaved species were concentrated and separated using size exclusion chromatography (SEC) (Fig. 1b). The recombinant RNAP assemblies eluted in two peaks corresponding to apparent MW of 400 kDa and 175 kDa, respectively. SDS-PAGE and mass spectrometry analysis revealed that peak 1 contains all eight ASFV RNAP subunits (Fig. 1c, and Supplementary Table 1), while peak 2 contains the subcomplex vRPB2, 3-11, 9, and 10. The expression and purification method employed delivered a pure sample with a reproducible yield of typically 2.5 mg of core RNAP per litre of insect cell culture.

### Recombinant ASFV RNA polymerase is catalytically active

To test the catalytic activity of recombinant ASFV RNAP, we utilised nonspecific in vitro transcription assays where RNA polymerisation is measured by the incorporation of $[\alpha\text{-}^{32}P]$-UTP into RNA using activated calf thymus DNA as template, which is precipitated using cold trichloroacetic acid (TCA) and quantified by scintillation counting[20]. This assay is independent of promoter sequences, general initiation factors, or RNA primers and thus provides an unadulterated assessment of RNA polymerisation activity. We tested the fractions eluting from SEC and detected robust activity in peak 1 confirming that the active site of the complete recombinant ASFV RNAP is intact (Fig. 1d). The fractions in peak 2 were inactive, as predicted by the lack of the vRPB1 subunit that includes the catalytic aspartate triad in the NADFDGD motif which is evolutionary conserved in all archaeo-eukaryotic RNAPs, including ASFV. To further characterise the ASFV RNA polymerase we tested a range of pH values, ionic strengths, temperatures, divalent cations,

## Table 1 | Overview of ASFV RNAP subunits and comparison with RNAPII and VACV RNAP

| Subunit classification | ASFV Gene ID | ASFV RNAP | Yeast RNAPII | VACV RNAP |
|---|---|---|---|---|
| **Catalytic core** | NP1450L | vRPB1 | RPB1 | Rpo147 |
| | EP1242L | vRPB2 | RPB2 | Rpo132 |
| **Assembly platform** | H359L | vRPB3-11 | RPB3 | Rpo35 |
| | | | RPB11 | |
| | CP80R | vRPB10 | RPB10 | Rpo7 |
| | – | – | RPB12 | – |
| **Auxiliary function** | – | – | RPB4 | – |
| | D205R | vRPB5 | RPB5 | Rpo22 |
| | C147L | vRPB6 | RPB6 | Rpo19 |
| | D339L[a] | vRPB7 | RPB7 | Rpo18 |
| | – | – | RPB8 | – |
| | C105R | vRPB9 | RPB9 | – |
| | – | – | – | Rpo30 |

Summarised from Cackett et al. [13].
[a]The N-terminal part of ASFV D339L is homologous to RNAPII RPB7 and VACV Rpo18 stalk subunits.

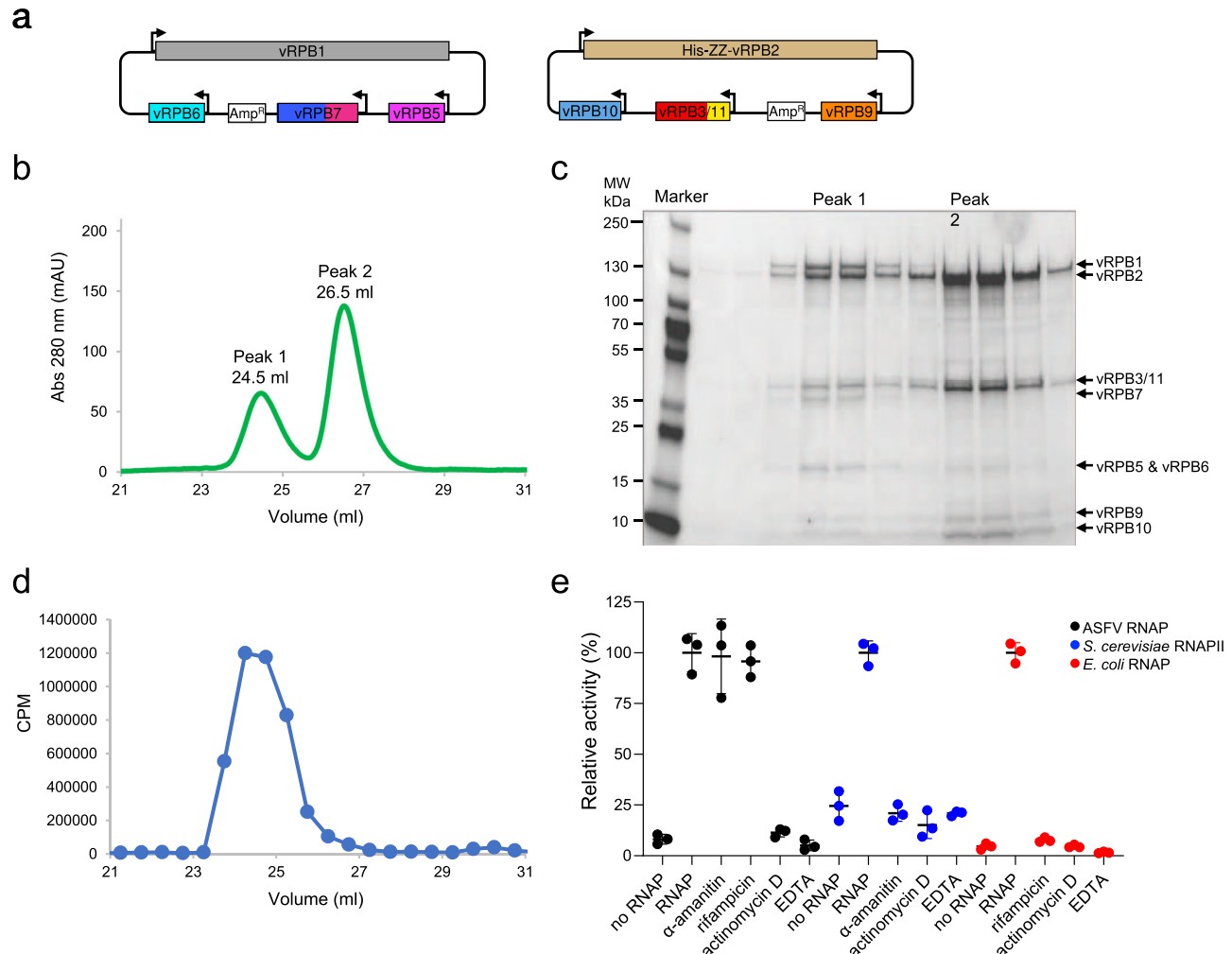

**Fig. 1 | Expression, purification, and biochemical characterisation of the ASFV RNAP. a** Overview of cloning strategy for the recombinant ASFV RNAP. The first bacmid-generating plasmid includes vRPB1 (grey), vRPB5 (magenta), vRPB6 (cyan), and vRPB7 (blue) with the CTD in deep pink. The second plasmid contains vRPB2 (tan), vRPB3-11 fusion (red and yellow, respectively), vRPB10 (cornflower blue) and vRPB9 (orange). vRPB2 includes an N-terminal cleavable ZZ-affinity tag. **b** UV profile of SEC purification. **c** SDS-PAGE analysis of SEC purification step reveals that the two peaks contain the complete RNAP (peak 1) and a subassembly (peak 2) lacking vRPB1, 5, 6 and 7, respectively. This is a representative gel, which was repeated six times. Source data are provided as a Source Data file. **d** Nonspecific in vitro transcription assay shows that only SEC peak 1 fractions contain robust transcription activity. Activity is reported as CPM (counts per minute) of radiolabelled [α−32P]-UTP incorporation as a function of the SEC elution volume (from panel b). The experiment was carried out only for the first purification. **e** Effect of RNAP inhibitors on ASFV core RNAP. Alpha-amanitin and rifampicin, both at 100 μM, do not inhibit the ASFV RNAP, while inhibition was observed for the DNA intercalator Actinomycin D (100 μM) and magnesium chelator EDTA (50 mM). Control experiments using *S. cerevisiae* RNAPII and *E. coli* RNAP were used to confirm alpha-amanitin and rifampicin inhibition. For all polymerases a negative control with only buffer (no RNAP lane) was prepared. Transcription activity is shown as % of radiolabelled [α−32P]-UTP incorporation relative to each RNA polymerase in the absence of inhibitors (labelled as RNAP) versus the same reaction without enzyme (no RNAP). Results were produced in triplicates and reported as the mean with corresponding standard deviation. Source data are provided as a Source Data file.

and different DNA templates. The results showed that the ASFV RNAP has an optimum at a temperature between 30 and 40 °C and pH 8.0 (Supplementary Fig. 1a, b). It is interesting to point out that the virus replicates in two very different hosts, in pigs which have a body temperature of 39 °C, and in soft ticks whose body temperature fluctuates with the environment. ASFV RNAP is sensitive to ionic strength in the nonspecific assay, being most active using low KCl concentrations (Supplementary Fig. 1c) and has a clear preference for magnesium over manganese (Supplementary Fig. 1d and 1e, respectively). The results obtained with the recombinant core RNAP expressed in insect cells are in good agreement with ASFV RNAP preparations isolated from virions[19]. Like other RNAPs[23,24], the activity of ASFV RNAP in a non-specific assay is higher with single stranded compared to double-stranded DNA (Supplementary Fig. 1f). In conclusion, we decided to

use assay conditions for the ASFV RNAP at 37 °C, pH 8.0, 5 mM MgCl$_2$, and using the more physiologically relevant double stranded DNA. Under these conditions, the specific activity of the ASFV core RNAP is 83 (±8) nmol incorporated UMP per hour per mg RNAP. This is comparable to *Saccharomyces cerevisiae* RNAPII, 20 (±2) nmol incorporated UMP per hour per mg RNAP, and the *E. coli* core RNAP, 130 (±7) nmol incorporated UMP per hour per mg RNAP obtained using our assay conditions, and also in agreement with results from the literature including the recombinant 12-subunit archaeal RNAP from *Methanocaldococcus jannaschii* (160 nmol incorporated UMP per hour per mg) measured under similar assay conditions[20].

We used this assay to test the sensitivity of ASFV RNAP to eukaryotic and bacterial RNAP inhibitors (Fig. 1e). While the metal chelator EDTA and the DNA-intercalating drug Actinomycin D reduced ASFV

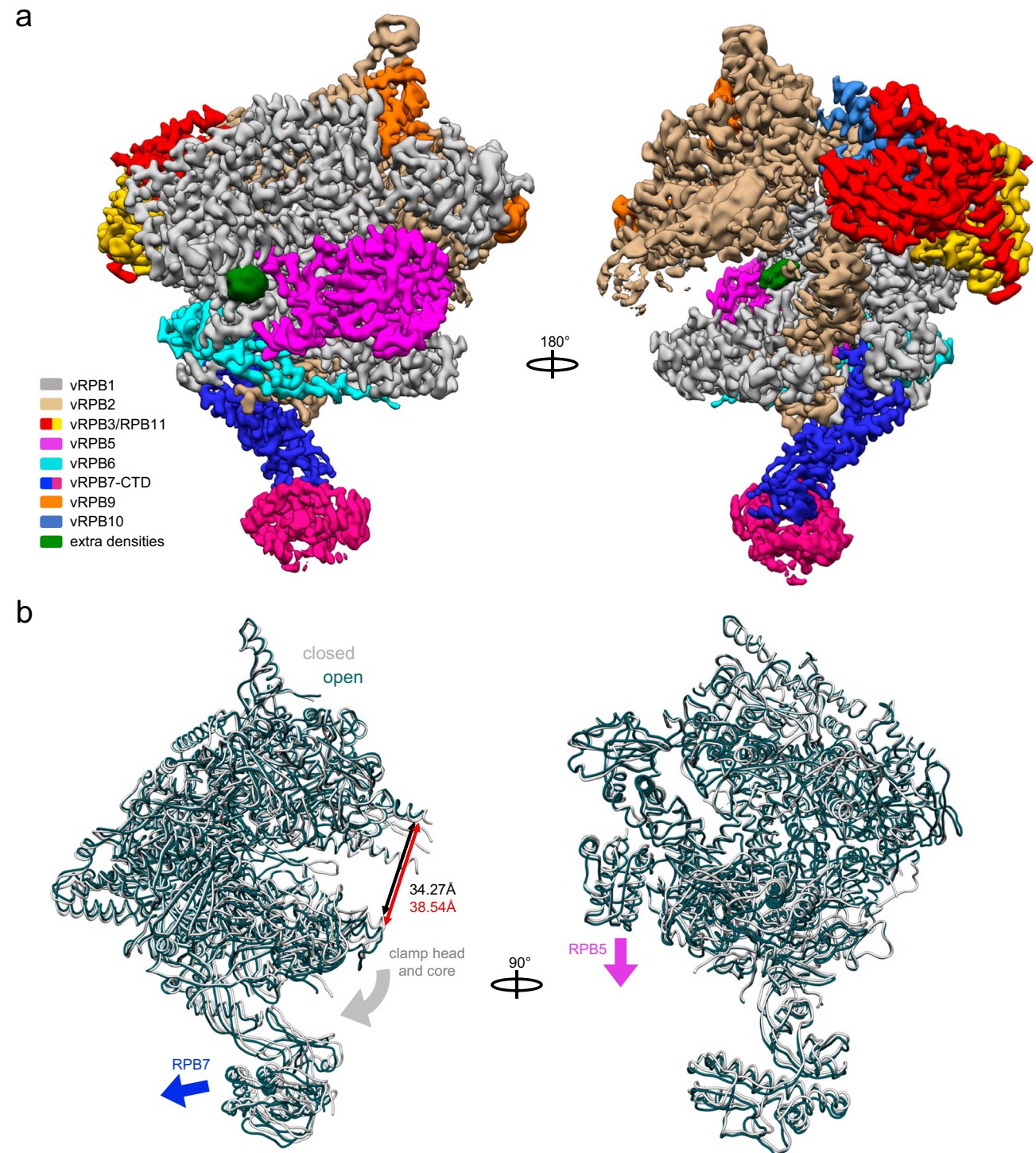

**Fig. 2 | The cryo-EM structures of the ASFV core RNA polymerase. a** The cryo-EM map of the 8-subunit RNAP with the subunits colour coded according to the legend. The EM map reveals two unidentified densities/ligands, shown in green. **b** Superposition of the structures corresponding to the closed and open conformations of the RNAP (grey and teal, respectively). Conformational changes involving movements of vRPB5 (magenta block arrow), vRPB7 (blue block arrow), as well as clamp head and core (grey block arrow) lead to the widening of the DNA-binding channel by 4.3 Å. The two conformations are shown in worm style and superimposed on the vRPB2 subunit. The width of the DNA-binding channel was measured between residues vRPB2 Val357 and vRPB1 Leu254 in Chimera v1.16.0[83].

RNAP transcription activity to background levels, neither high concentrations (100 μM) of alpha-amanitin nor rifampicin affected ASFV RNAP activity. These results are in good agreement with experiments carried out in 1980 using transcription-competent extracts obtained from ASFV particles[19]. Having ascertained the integrity of recombinant ASFV RNAP in terms of subunit composition and catalytic activity, we solved the RNAP structure using cryo-EM.

## The ASFV RNAP structure resolved by cryo-EM

The molecular structure of the 8-subunit core ASFV RNA polymerase was solved by imaging the sample under native conditions (Fig. 2 and Supplementary Fig. 2). The data processing of 3,825,942 high quality particles, extracted from 14,638 movie-stacks, highlighted from the very beginning the presence of multiple conformations that we addressed by implementing a 3D variability analysis (3DVA)[25]. From

**Table 2 | Cryo-EM data collection, refinement and validation statistics**

| | Closed conformation (EMDB-18120) (PDB 8Q3B) | Stalk[a] (EMDB-18264) | Open conformation[a] (EMD-18163) | Open conformation (composite map)[a] (EMDB-18129) (PDB 8Q3K) |
|---|---|---|---|---|
| **Data collection and processing[b]** | | | | |
| Magnification | 105,000 | - | - | - |
| Voltage (kV) | 300 | - | - | - |
| Electron exposure (e-/Å$^2$) | 48.152 | - | - | - |
| Defocus range (μm) | 1.5 - 2.7 | - | - | - |
| Pixel size (Å) | 0.828 | - | - | - |
| Symmetry imposed | C1 | - | - | - |
| Initial particle images (no.) | 4,842,857 | - | - | - |
| Final particle images (no.) | 467,000 | 467.519 | 352,192 | n/a |
| Map resolution (Å) | 2.69 | 2.99 | 2.92 | n/a |
| FSC threshold | 0.143 | 0.143 | 0.143 | |
| Map resolution range (Å) | n/a[c] | 2.84 - 6.55 | 2.82 - 5.72 | n/a |
| **Refinement** | | | | |
| Initial model used (PDB code) | none | - | - | 8Q3B |
| Model resolution (Å) | 2.9 | - | - | 3.0 |
| FSC threshold | 0.143 | | | 0.143 |
| Model resolution range (Å) | 2.7 - 3.0 | - | | 2.8 - 3.2 |
| Map sharpening $B$ factor (Å$^2$) | 113.15 | 70.28 | 84.87 | n/a |
| Model composition | | - | - | |
| Protein residues | 3724 | | | 3633 |
| Ligands | 1 Mg, 6 Zn | | | 1 Mg, 6 Zn |
| $B$ factors (Å$^2$) | | - | - | |
| Protein | 83.16 | | | 94.15 |
| Ligand | 140.13 | | | 174.54 |
| R.m.s. deviations | | - | - | |
| Bond lengths (Å) | 0.003 | | | 0.003 |
| Bond angles (°) | 0.552 | | | 0.511 |
| Validation | | - | - | |
| MolProbity score | 1.23 | | | 1.29 |
| Clashscore | 4.62 | | | 5.45 |
| Poor rotamers (%) | 0.46 | | | 0.22 |
| Ramachandran plot | | - | - | |
| Favoured (%) | 98.05 | | | 98.05 |
| Allowed (%) | 1.95 | | | 1.95 |
| Disallowed (%) | 0.00 | | | 0.00 |

[a]The stalk (EMDB-18264) was obtained from the multibody refinement of the closed conformation and combined with the open conformation (EMDB-18163) to produce the composite map (EMDB-18129). Only the latter was used for the refinement.

[b]All maps were obtained from the same dataset.

[c]The local resolution evaluation was carried out in cryoSPARC which does not provide this information.

this analysis it emerged the dataset could be described by three key conformational changes, (i) a widening of the DNA-binding channel up to 4.3 Å (Supplementary Movie 1, and Fig. 2b), (ii) the swivelling of the shelf module (Supplementary Movie 2) that has been associated with pausing and termination in bacterial RNAP[26], and (iii) the flexibility of the vRPB7 stalk (Supplementary Movie 3). Based on the output of the 3DVA, as well as the extent and relevance of the movements, we focused on two structural states that are characteristic for DPBB RNAPs, corresponding to the closed and the open RNAP DNA-binding channel conformations of ASFV RNAP. Subsequently, the batches of corresponding particles were isolated and provided two maps at the nominal resolution of 2.73 Å for the closed conformation, and 2.92 Å for the open conformation (Table 2, and Supplementary Fig. 3). To improve the map quality surrounding the intrinsically flexible stalk, we successfully applied a 3D multibody refinement[27] resulting in a 2.99 Å resolution map (Supplementary Fig. 3d, e). A structural model for each

RNAP subunit was generated by AlphaFold2[28], each displaying a high structural similarity with their RNAPII counterparts. The high resolution achieved allowed us to confidently fit the predicted models and manually edit the less accurately predicted regions, associated in most cases with flexible loops, and domains such as the protrusion and the wall in vRPB2. However, to achieve better structural information for the protrusion and wall domains, we employed a new refinement method, named 3DFlex refinement[29], applied after 3D refinement of the closed conformation map. The new refinement produced a higher quality map, improving the local resolution in particular of mobile domains exposed on the surface like vRPB5, vRPB7, whose map quality was comparable to the multibody refinement result, and the protrusion (for comparison see Supplementary Figs. 3a and 4a). In good agreement with predictions based on subunit sequence, the ASFV RNAP includes six zinc finger motifs, two in each vRPB1 and vRPB9, one in vRPB2 and in vRPB10, as highlighted in Supplementary Figs. 5c, 6,

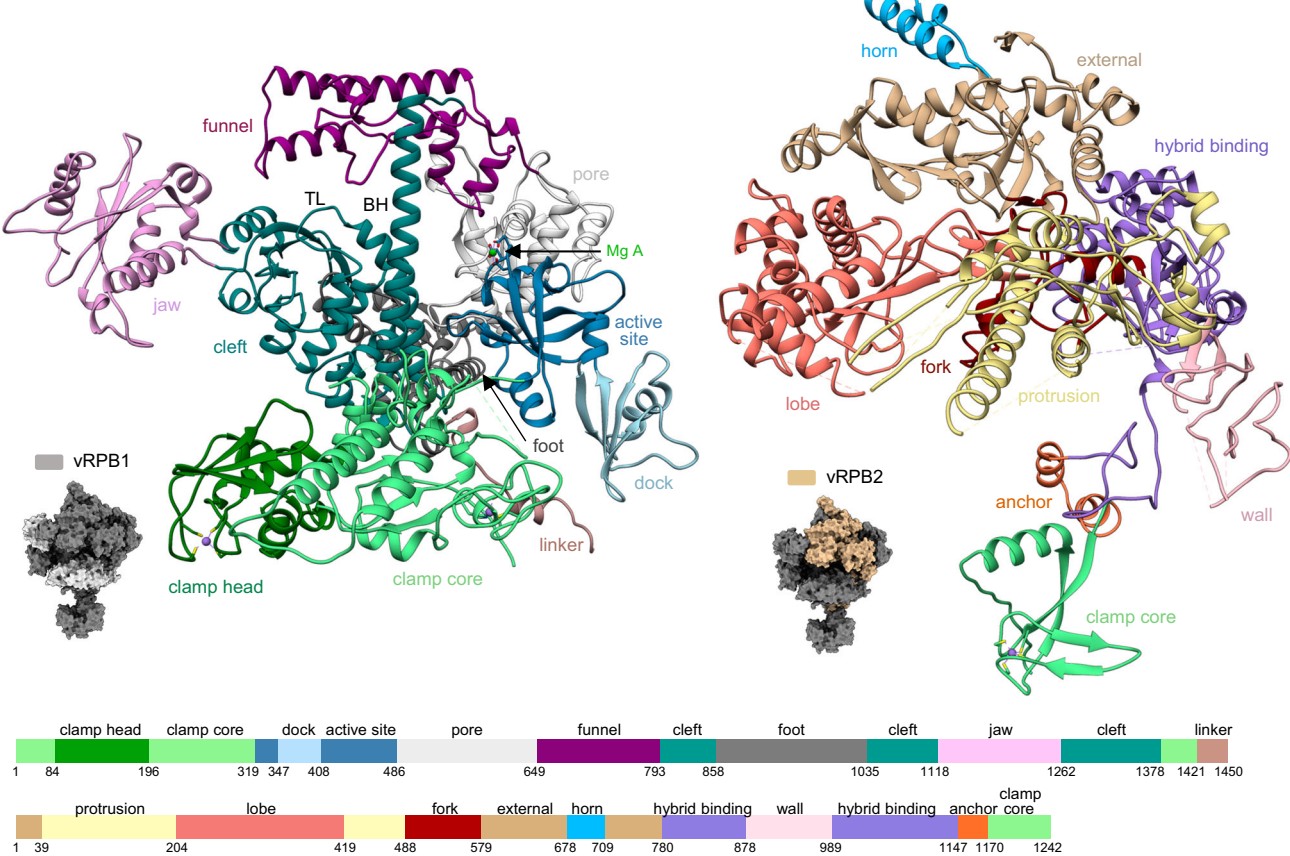

**Fig. 3 | Features of the large ASFV RNA polymerase subunits.** The location of the large subunits vRPB1 (light grey) and vRPB2 (beige) is shown in the context of the RNAP shown in a surface representation (dark grey). The structures of vRPB1 and vRPB2 subunits are shown in ribbon style and the domains highlighted in different colours according to the domain organisation below. Zinc ions are highlighted in medium purple and magnesium in green. The bridge helix and the trigger loop are labelled as BH and TL, respectively.

and 7. In addition, two proline residues were found in the cis configuration, both in vRPB1 at positions 374 and 421 (Supplementary Fig. 5b).

We identified two additional densities of unknown nature in both maps of the closed and the open RNAP (Fig. 2a). The first density is located inside the DNA-binding channel, forming a stacking interaction with the side chain of residue Y824 on the bridge helix (BH) and surrounded by several positively charged residues (Supplementary Fig. 5d). The second density is located at the foot domain of vRPB1, and also surrounded by positively charged residues (Supplementary Fig. 5e). Whether these ligands are physiologically relevant and affect RNAP activity or its conformational flexibility remains unknown.

**The ASFV RNAP structure is related to RNAPII**
The main body of ASFV RNAP is comprised of the vRPB1 and vRPB2 subunits that contain all conserved domains and structural motifs characteristic for cellular DPBB RNAPs[1] (Fig. 3 and Supplementary Figs. 6, and 8). The active centre is well resolved in both the closed and the open RNAP conformation maps showing the catalytic magnesium A ion (Mg A) fully coordinated by the aspartate triad D457, D459, and D461 in the conserved NADFDGD motif (Supplementary Figs. 5a and 6). The bridge helix (BH) is in the straight conformation, while the trigger loop (TL) is structurally well-defined in both closed and the open ASFV RNAP structures. The assembly platform of ASFV RNAP is formed by the fused vRPB3 and 11 polypeptide and vRPB10. This composition is conserved in some NCLDV RNAPs including VACV[6,30], however, the ASFV vRPB3-11 fusion preserves the structural organisation and domain orientation of RNAPII (Fig. 4a and Supplementary

Figs. 7 and 9), while in VACV the low sequence conservation coupled with a fusion event has led to a different orientation and domain organisation for RPB11. Lastly, eukaryotic RPB3 homologues contain a zinc finger domain, and some archaeal Rpo3 subunits also an iron-sulphur cluster, while the VACV and ASFV vRPB3-11 have no metal centres. RPB12 is shared between RNAPI, II and III, but not widely distributed among NCLDVs[30]. In ASFV, the loss of RPB12 is not replaced by any additional moiety provided by vRPB2 or vRPB3-11, which leads to a greater flexibility of the ASFV vRPB2 wall. As result of the flexibility, we could not resolve the wall domain in its entirety (Fig. 3 and Supplementary Figs. 4a and 8). In addition, we observed a loss of map density for the vRPB2 protrusion (Fig. 2a). The protrusion domain is known to be flexible, however in ASFV the degree of flexibility seems to be unusually higher. At sequence level (Supplementary Fig. 6), the protrusion is longer compared to RNAPII which might lead to larger movements and in part explain the loss of density.

A hallmark feature of archaeal and eukaryotic DPBB RNAPs is the heterodimeric Rpo4/7 and RPB4/7 (in archaea and RNAPII, respectively) stalk that protrudes from the crab claw-like shape of the enzyme[31]. The stalk is conformationally flexible and likely to interact with the nascent RNA via the OB-fold domain in RPB7[32] and Rpo7[33]. The ASFV RNAP stalk is comprised of only one polypeptide, vRPB7, which consists of an N-terminal domain that is structurally conserved with RPB7, and a C-terminal domain (CTD) with no sequence or known structural homology, that occupies the space of the missing RPB4 subunit (Fig. 4b, and Supplementary Figs. 7 and 9).

Finally, the auxiliary subunits vRPB5, vRPB6, and vRPB9 are all closely related to their RNAPII counterparts (Fig. 4c, d, and

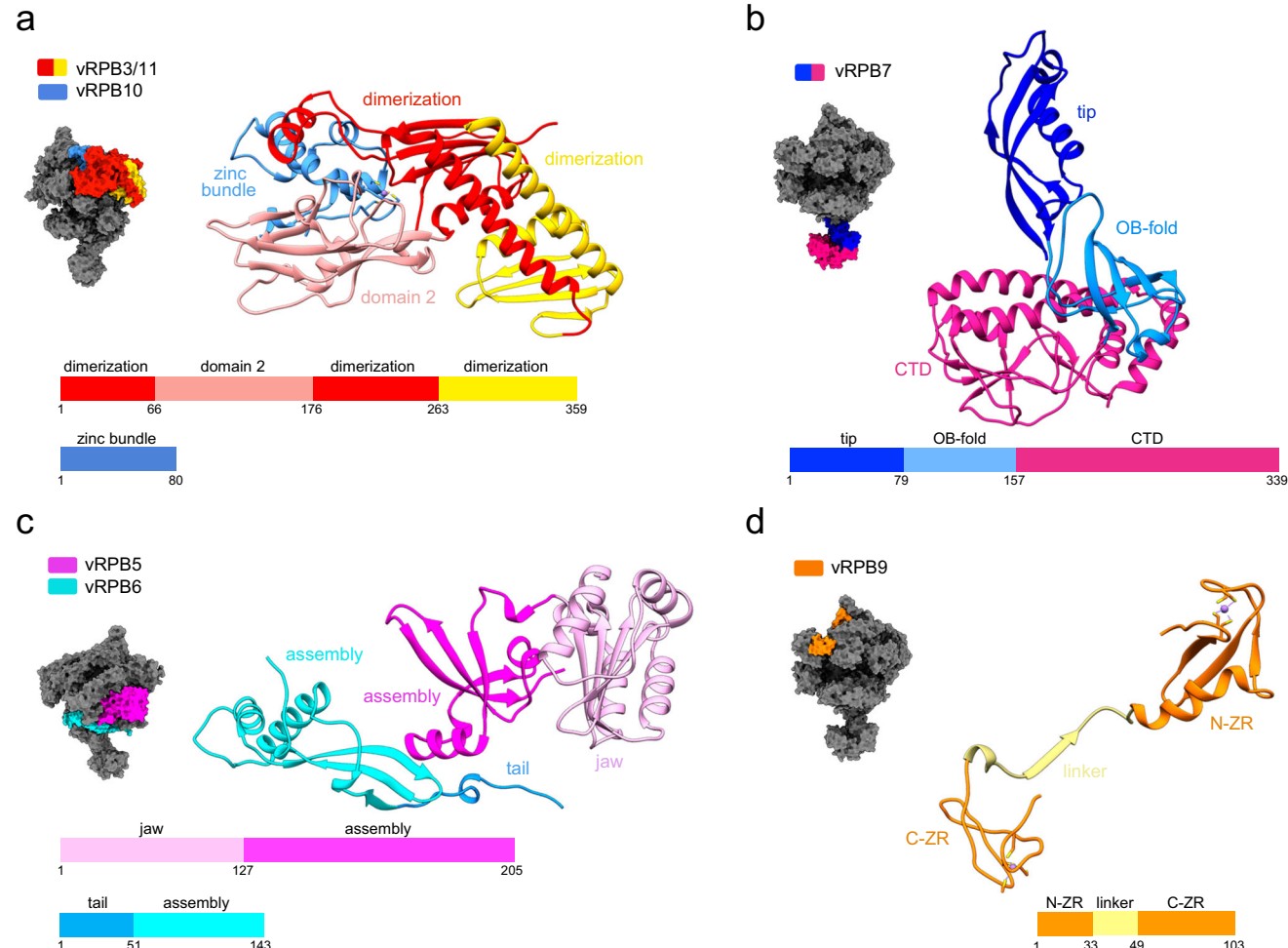

**Fig. 4 | Features of the small ASFV RNA polymerase subunits.** The location of the colour coded small subunits is shown in the context of the RNAP in a surface representation (dark grey), while each subunit is shown in ribbon style with domains coloured according to the schematic reported below each structure. Zinc ions are highlighted in medium purple. **a** The assembly platform consists of a single fused vRPB3-11 polypeptide and vRPB10. **b** The vRPB7 stalk subunit has a large virus-specific CTD (deep pink). **c** vRPB5 and vRPB6. **d** vRPB9 is comprised of two zinc ribbon domains named N-ZR for the N-terminal domain and C-ZR for the C-terminal one connected by a linker (pale yellow).

Supplementary Figs. 7 and 9). vRPB5 has a bipartite structure that includes the N-terminal assembly domain, that binds to vRPB1 and is conserved in both archaea and eukaryotes, and the eukaryote-specific C-terminal RNAP jaw domain which interacts with the downstream duplex DNA. vRPB6 makes conserved contacts with vRPB1 and likely enhances RNAP assembly and/or stability like RPB6, Rpo6 and ω in eukaryotes, archaea and bacteria, respectively[34]. vRPB9 consists of two zinc-ribbon domains connected by a flexible linker, where both the structures and binding sites on the RNAP are strictly conserved with RNAPII.

## Unique structural features of the ASFV RNAP

Given the similarity of the ASFV RNAP with other DPBB RNAPs, and to highlight relevant structural features unique to ASFV, we prepared structural alignments and superimposed the ASFV RNAP with human RNAPII and VACV RNAP (Supplementary Figs. 6–9). Apart from the above-mentioned differences in RNAP subunit composition (RPB4, 8, and 12 missing in ASFV) and protein fusions (RPB3-11) that do not alter the structure of the conserved RNAP subunits, we focussed on small-scale alterations, such as missing or additional domains and motifs which could affect interactions with virus-specific or host-related transcription factors, and with nucleic acids. One important difference concerns the NTP-entry channel, that enables the access of regulatory

factors and NTPs to the active site. The vRPB1 rim helices in the funnel of the NTP-entry channel are generally not highly conserved on the sequence level, but always form two continuous alpha-helices connected by a loop of variable length depending on the species. Specifically, the C-terminal rim helix is directly involved in binding the C-terminal domain of eukaryotic transcript cleavage factor TFIIS[35], its paralogous RNAPI RPA12[36], and RNAPIII RPC11[37], as well as the archaeal factors TFS1[38] and TFS4[39]. Importantly, this is also the binding site of the fungal RNAPII inhibitor alpha-amanitin[40]. In ASFV RNAP, the folding of the C-terminal rim helix is broken between residues L709 and D713 (Supplementary Figs. 6 and 8). This folding impairment, most likely due to residues F711 and P712, unique to ASFV, disrupts the alpha-amanitin binding site. This alteration provides the structural basis for the resistance of the ASFV RNAP to alpha-amanitin evidenced in the in vitro transcription assays (Fig. 1e). ASFV encodes a homologue of TFIIS, vTFIIS[41], but its interaction with RNAP has not yet been investigated.

Superimposition of the initially transcribing complexes of the human RNAPII[42] and VACV RNAP[15] with the ASFV RNAP structure allowed a closer scrutiny of the active site environment and revealed small differences in the fork loop 2 and helix (residue 794 - 801) motifs in vRPB2 (Supplementary Fig. 10). The fork loop 2 in ASFV is replaced by the less flexible alpha-helices connected by a short loop; however,

the position of the loop is perfectly overlapping with the fork loop 2 in RNAPII. By contrast, the vRPB2 helix 794 - 801 runs perpendicularly towards the RNA binding site, instead of forming a stable interaction with the rest of the hybrid binding domain as seen in VACV and human RNAPII. Yet, none of the two motifs seem to interfere with the scaffold modelling. In addition, the vRPB2 subunit contains a unique motif comprised of 31 amino acids (residue 679 to 709) located in the external domain (Fig. 3 and Supplementary Fig. 6). This motif, which we coined the horn, protrudes from the RNAP surface by forming an alpha-helix that is stabilised by interacting with the C-terminal zinc ribbon domain of vRPB9, and folds back with a second short helix followed by a linear unstructured segment. The horn is not present in any other cellular or viral RNAP characterised thus far. The position of the horn, and its large exposed surface area, suggests it serves as a binding site of a not yet identified virus-specific factor.

### The vRPB7 CTD as a potential docking site for the capping enzyme

The most unusual feature of the ASFV RNA polymerase is the 175 amino acid long C-terminal domain (CTD) of vRPB7. This domain is highly conserved in NCLDVs closely-related to ASFV (Supplementary Fig. 11a, b) but absent in most NCLDV families, including poxviruses. To identify the possible evolutionary origin and potential function of this domain we carried out a structural homology search using the DALI webserver[43]. The results identified several viable candidates with a relatively low structural conservation, which is not unusual for viral proteins (Supplementary Table 2). Most hits covered less than the half of the vRPB7 CTD with two significant exceptions that account for more than two thirds of the domain, both related to the mRNA capping machinery: (i) the human mRNA cap 2´O-MTase 1[44] scored the top of the DALI search despite a low 8% sequence identity, and (ii) the VACV capping enzyme accessory subunit D12[7] albeit with an even lower sequence identity of 3% within the equivalent structural regions. A superimposition and structural alignment (Fig. 5a) of the 2´O-MTase 1 with either vRPB7 (Fig. 5b) or D12 (Fig. 5c) clearly shows the similarity. While the first alpha-beta structural modules characteristic for MTases are conserved in both ASFV vRPB7 CTD and VACV D12, the last alpha-helical layer that binds the S-adenosyl methionine (SAM) cofactor and the mRNA substrate is missing in both viral domains. In VACV, D12 interacts with the N7-MTase domain of the capping enzyme D1 subunit[6,7]. But in contrast to VACV D12, the ASFV MTase-like domain appears to be stably integrated into the RNAP at the C-terminus of vRPB7. The Alphafold2 modelling of additional NCLDV RPB7 paralogs from Abalone, Fausto-, Kaumoeba-, and Pacmanviruses supports the possible conservation of this in-built putative CE subunit, specifically amongst *Asfarviridae*-related NCLDVs (Supplementary Fig. 11c). Using the published X-ray structure of the ASFV CE N7-MTase domain (pdb 7d8u)[45] with our cryoEM structure of ASFV RNAP, and by using the VACV D1-D12 interface as a guide[7], we prepared a model of the predicted complex formed by the ASFV RNAP and N7-MTase CE domain (Fig. 5d). The CE N7-MTase domain fits snugly alongside vRPB7 without any major steric clashes, supporting a model where the ASFV CE is stably incorporated into the RNAP rather than reversibly associate-dissociate during the transcription cycle akin to nuclear RNAPII or in VACV.

## Discussion

Due to the combination of high infection and mortality rates and the absence of approved vaccines and antiviral drugs, ASFV has a devastating impact on the global food system that comes at a large cost to society with damages estimated at USD 100+bn[46]. The scarcity and increased cost of pork products in turn motivates people to search for alternative food sources in the wild which has been associated with zoonosis of other animal viruses like Ebola virus and SARS-CoV2[47,48]. As RNAPs are essential, they provide ideal targets for therapeutic drugs to

fight pathogens. Ultimately, vaccines are required to eradicate ASF, but antiviral agents can play an important role to contain local outbreaks and prevent further spreading of the disease, as it has been modelled with classical swine fever[49]. Recent work has shown targeting of ASFV transcription to be a promising strategy in the context of inhibiting ASFV in infected or gene-transfected tissue culture cells[50]. However, the identification of such antivirals against ASFV RNAP is greatly facilitated by (i) transcription activity-based in vitro assays suitable for screening RNAP inhibitors, and (ii) the molecular structure of the pathogen's RNAP at high-resolution, which is required for a rational structure-based drug design approach. In the current work, we provide both.

We have reported the efficient preparation of recombinant ASFV core RNAP in insect cells at a suitable milligram-scale and a nonspecific in vitro transcription activity assay which is independent of promoter templates and therefore suitable to screen for inhibitors of transcription during early and late stages of infection (Fig. 1 and Supplementary Fig. 1). We have furthermore solved the structure of the ASFV core RNAP at 2.7-2.9 Å resolution (Fig. 2 and Table 2) that is required for a structure-based in silico drug design. The ASFV core RNAP emerges as a close *replica* of the virus' host RNAPII. The ASFV RNAP adopts the prototypical crab claw-like shape of cellular RNAPs formed by the two large subunits vRPB1 and 2, which contribute one DPBB each to the catalytic centre at their interface (Fig. 3). Despite an overall good congruence with the structure of other metazoan RNAPII, the low amino acid sequence identity and the local structural divergence render the ASFV RNAP resistant to alpha-amanitin (Fig. 1e and Supplementary Fig. 8). The differences between cellular and ASFV RNAPs in turn improve the feasibility that drugs can be developed with sufficient specificity and selectivity to circumvent cross-reaction with host transcription machinery. Using our in vitro transcription assay, small molecule libraries, including NTP analogues, can now be screened to identify lead compounds for the development of antiviral drugs. Cellular DPBB RNAPs cycle through an ensemble of structural states as the enzyme progresses through the transcription cycle, in response to DNA binding, association with initiation and elongation factors, and upon binding of negative regulators of RNAP. These include RIP and TFS4 which are involved in the virus-host arms race in archaea[39], Gfh1[51] and DksA[52] utilised in bacteria during non-optimal growth conditions, or eukaryotic regulators NELF1[53] and MAF1[54]. We solved the structure of the ASFV core RNAP in two conformations, i. e. an open and closed state, which refer to the mobile RNAP clamp and the width of the DNA-binding channel (Fig. 2b). The structures confirm that the structural flexibility inherent in viral RNAP is evolutionary conserved with cellular RNAPs.

The high degree of structural conservation allowed the modelling of the B-ribbon domain of the general initiation factor TFIIB and the DNA/RNA scaffold on the ASFV RNAP using both RNAPII and VACV RNAP structures as templates (Fig. 6). Factors related to TFIIB are essential for transcription initiation of the archaeal RNAP (TFB)[55], eukaryotic RNAPI (TAF1B)[56], II (TFIIB)[57], and III (Brf1)[58], as well as VACV RNAP (Rap94)[6]. Specifically, interactions between the B-ribbon of TFIIB-like factors and the RNAP dock domain are instrumental for RNAP recruitment to promoters. The TFIIB homolog of ASFV is encoded in the viral genome and thought to facilitate late, and possibly intermediate, gene transcription[13,17]. Both human TFIIB and VACV Rap94 B-ribbon and B-reader domains fit well in the ASFV RNAP structure suggesting that the molecular mechanisms of TFIIB/RNAP recognition during initiation is conserved in ASFV.

It has been hypothesised that NCLDV RNAPs are derived from a proto-eukaryotic, archaea-like host that existed prior to the last eukaryotic common ancestor (LECA), and it has been furthermore speculated that viral RNAP subunits later were transferred from viruses back to eukaryotes after undergoing sequence diversification to enable the diversification of RNAP into the three main classes of RNAPI,

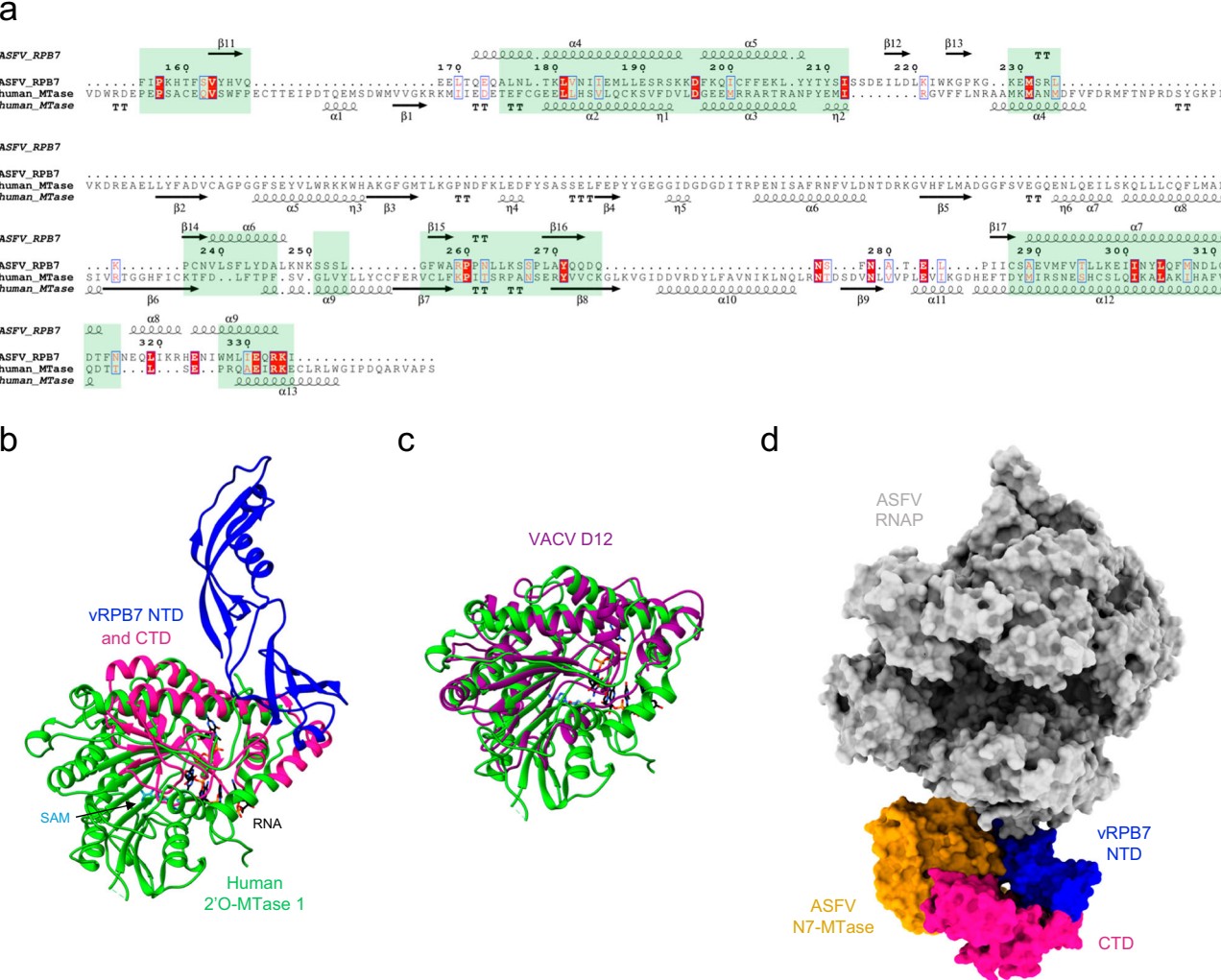

**Fig. 5 | The RPB7 CTD is related to the 2´O-MTase domain. a** Structure-based sequence alignment generated after superimposition of the ASFV vRPB7 CTD and the human 2´O-MTase 1 (pdb 4n48[44]). The alignment was rendered using Esprit 3 webserver[89], applying an equivalence score of 0.6 (range 0 to 1) and including a description of the secondary structure composition of both proteins, based on the provided structures (according to the default Esprit 3 labelling method). The structural equivalences resulted from the DALI search are highlighted within light green boxes. **b** Superimposition of the cryoEM structure of ASFV vRPB7 (NTD in blue and CTD in deep pink) and the crystal structure of the human 2´O-MTase 1 (in green). The SAM cofactor (sky blue) and GTP-capped RNA substrate (black) bound to the 2´O-MTase 1 are shown in stick representation. **c** Superimposition of the human 2´O-MTase 1 and the VACV D12 CE subunit (dark magenta, pdb 6rie[7]). **d** Model of the ASFV RNAP-CE in a surface representation. The ASFV RNAP is shown in light grey, with the exception of vRPB7 NTD (blue) and CTD (deep pink). The crystal structure of the ASFV N7-MTase (pdb 7d8u dark gold) was docked onto the ASFV RNAP using the subunits interface of VACV D1-D12 as reference.

II and III in extant metazoans[59]. This raises the question which archaeo-eukaryotic RNAP subunits are missing in ASFV, and what that tells us about the selective pressures that have shaped the viral RNAP. The assembly platform has been simplified from four to two polypeptides, the fused vRPB3-11 and vRPB10, while RPB12 is lacking (Fig. 4a), streamlining RNAP assembly. RPB12/Rpo12 is located proximally to TBP in eukaryotic and archaeal preinitiation complexes (PIC), and Rpo12 is involved in transcription initiation[60,61]. Considering ASFV encodes initiation factors homologous to both archaeo-eukaryotic TBP[62] and VACV A7[63] (which includes a TBP-like domain[15]), it is likely that they interact with ASFV RNAP in a different, RPB12/Rpo12-independent manner. The subunit RPB8 is shared between RNAPI, II and III and has been suggested to facilitate the import of RNAPs across the nuclear envelope[64,65], a function that is irrelevant to cytoplasmic NCLDVs like ASFV. RPB8 emerged in archaea, prior to eukaryogenesis, however, it has a variable phylogenetic distribution in archaea[66,67].

The stalk domain of archaeal and eukaryotic RNAPs is conformationally flexible and interacts with the nascent RNA via the OB-fold domain in RPB7[32] and archaeal Rpo7[31,33]. This OB-fold is related to

the bacterial NusA factor S1 domain[68] and similar to *Escherichia coli* NusA, Rpo7 mutations that interfere with RNA binding also impair elongation processivity and termination efficiency in archaea[69]. In addition to facilitating mRNA interactions, the stalk in eukaryotes serves to couple transcription and mRNA processing. It recruits 5´-mRNA processing factors, including the CE[70], and coordinates transcript cleavage and polyadenylation responsible for mRNA 3´-end formation[71]. ASFV RNAP has a clearly defined stalk comprised of vRPB7 which includes an OB-fold domain but lacks the archaeo-eukaryotic Rpo4/RPB4 subunit (Figs. 2, and 4b). While RPB7 and Rpo7 are essential for cell viability in eukaryotes and archaea, respectively, RPB4/Rpo4 is not essential[72], which at least in part rationalises why the ASFV RNAP can work without RPB4. Instead, vRPB7 is C-terminally fused to a small domain with structural homology to the 2´O-MTase 1 subunit of the human CE[44], and to the VACV capping enzyme D12 subunit[7] (Fig. 5). Akin to D12, the vRPB7 CTD is vestigial and has lost the cofactor (SAM), and the RNA-binding domain, however, it could still serve as interaction partner to facilitate the recruitment of the CE, i.e. forming an in-built CE subunit. The ASFV CE is homologous

a

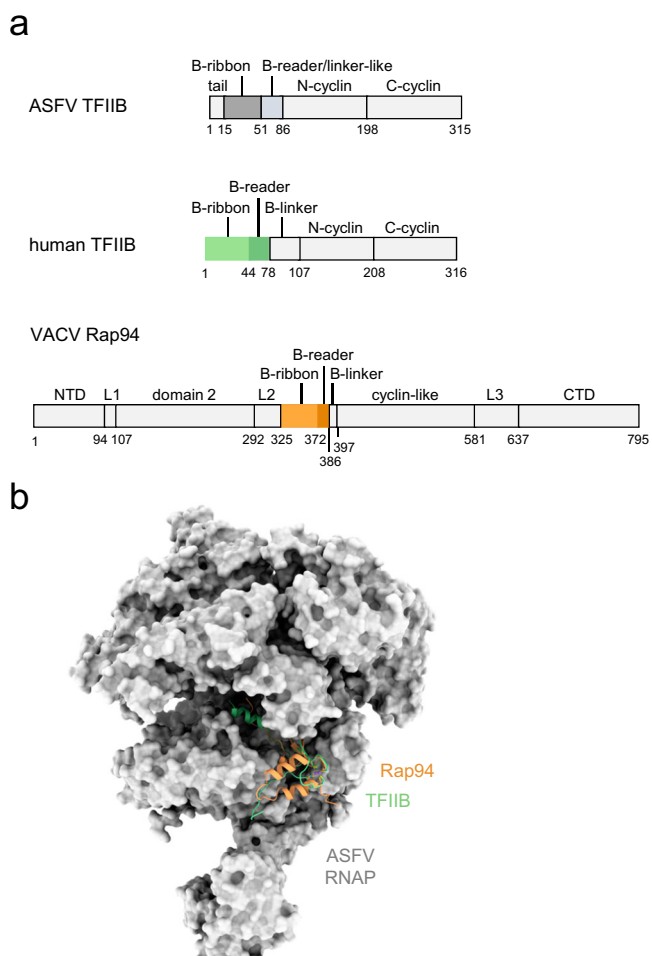

b

**Fig. 6 | Modelling of the B-ribbon domains into the ASFV RNAP structure. a** The domains organisation of ASFV and human initiation factor TFIIB, as well as of the distant homologous VACV Rap94, are shown in a gradient of grey with the exception of the B-ribbon and B-reader domains that are shown in the structure in panel (b). In ASFV TFIIB, the B-reader and B-linker motifs are shorter than in human TFIIB and, without a structure available, by sequence conservation it is not possible to establish whether they are either both present or compromised. Thus, they are shown in the figure as a single domain and labelled as B-reader/linker-like. **b** The human RNAPII/TFIIB complex (pdb 3k1f[57]) and VACV preinitiation complex (PIC, pdb 6rfl[6]) were superimposed to the ASFV closed conformation structure, shown in grey surface representation. For both RNAPII/TFIIB and VACV PIC, only the B-ribbon and the B-reader are shown in light green and light orange ribbons style, respectively. All superpositions were prepared in Chimera[83] using vRPB2 as reference, while the surface rendering was prepared in Chimera X[90].

to the VACV D1 subunit, while ASFV does not encode obvious homologues of the VACV D12 subunit[12] (Supplementary Fig. 12a). VACV D12 binds the D1 subunit N7-MTase domain, with D12 acting as an allosteric activator for N7 methylation of the guanine[73,74]. The human 2´O-MTase 1 methylates the 2´O ribose of the first mRNA residue[44] and it interacts directly with both the RNAPII stalk and the capping enzyme that only encodes the TPase and GTase domains in the human system[70]. This supports the scenario in which vRPB7 CTD eases recruitment and retention of its CE and subsequently co-transcriptional capping. The feasibility of vRPB7-CTD/CE complex formation is supported by a docking model of our ASFV RNAP and the CE N7-MTase domain (Fig. 5d). This proposed model, however, needs to be validated by structural characterisation of the native ASFV RNAP/CE complex. While the domain organisation of the VACV CE D1 subunit and ASFV CE is conserved (TPase, GTase, OB-fold, and N7-MTase), the ASFV RNAP/CE model differs from VACV co-transcriptional capping complex

(CCC) because the ASFV CE N7-MTase domain is predicted to be on the other side of vRBP7 due to the hypothetical interaction with the vRPB7 CTD (Supplementary Fig. 12b, and c). However, we note that the topology of the RNAP/CE complex is highly variable in VACV, as the CE in the context of the VACV preinitiation complex (PIC) remains proximal to the RNAP stalk but in a different orientation and displaced from the RNAP core due to the binding of other virus-specific transcription factors (Supplementary Fig. 12d). ASFV, furthermore, encodes a polyadenylation polymerase, but whether this enzyme interacts with vRPB7 functionally like in eukaryotes[71] is unknown.

In conclusion, our work provides valuable insights into the structure, function, and evolution of the ASFV RNAP. Our recombinant system opens the doors for future studies with the power to unravel the detailed molecular mechanisms of ASFV transcription, including the temporal regulation of early and late viral gene expression. This is key to understanding the life cycle of ASFV and thought to be facilitated by interactions of the ASFV RNAP with distinct subsets of virus- and host-related initiation factors[13]. Since all early transcription factors are packaged into ASFV particles and due to the availability of structural information of PICs in the VACV system, we can make reasonable predictions about the factor requirements for early ASFV transcription, including roles for the ASFV homologues of VACV CE subunit D1, NPH-I and VETFs (D6) and VETFl (A7)[6,7,13]. Our understanding of the mechanisms of late gene control, the stage where most viral transcription occurs[17], is still lacking in ASFV, and insights into the structural basis and interactions between RNAP and late factors are currently non-existent. The recombinant ASFV RNAP system we present here now allows a rigorous analysis of viral transcription in vitro. Importantly, the tools we have developed provide promising opportunities for the search of antiviral drugs aimed at fighting ASFV and alleviating its threat to global food security.

## Methods

### Cloning of ASFV RNAP subunits into pLIB vectors

Our approach for expressing the eight-subunit recombinant ASFV RNAP, consisting of ASFV vRPB1, -2, -3, -5, -6, -7, -9, and -10 (genes NP1450L, EP1242L, H359L, D205R, C147L, D339L, C105R, and CP80R, respectively), utilised the biGBac system by Weissmann et al. [75]. Firstly, the wild-type ASFV-BA71V genes were synthesised in pUC57 plasmids by GenScript. Each gene was then Q5 PCR-amplified (NEB) and gel-extracted from pUC57 (Zymoclean Gel DNA Recovery Kit, Zymo Research), before Gibson Assembly[76] using HiFi NEBuilder (NEB) into PCR-linearised pLIB vectors from the biGBac system (primers described in Supplementary Table 3). RPB2 was assembled into a pLIB vector with an N-terminal TEV cleavable His-ZZ-tag, for downstream purification via ZZ-tag affinity. Genes inserted into pLIBs were checked via Sanger sequencing (Source Bioscience, Cambridge) before downstream steps.

### Stepwise Gibson Cloning into pBIG1 Vectors

Following assembly of each ASFV RNAP subunit gene into a pLIB vector, we used a two-step Gibson assembly method for generating four different pBIG1 plasmids, each containing two RNAP subunits. Throughout we will use the nomenclature of vectors and primers from the biGBac system by Weissmann et al., whose methods were followed with some changes, described here. As an example, we will go through the steps for generating pBIG1a with RPB1 and RPB6 inserted. The first step was to PCR amplify RPB1, with primers CasIF, and CasωR to produce the RPB1 gene expression cassette (GEC) fragment, generating overlaps to facilitate Gibson assembly with PmeI-digested pBIG1a to generate the pBIG1a:1 vector. pBIG1a:1 was then PCR-amplified with the CasIIF primer and our GAαR primer (Supplementary Table 3) to produce a linearised plasmid with an α and β overlap. Primers CasIF and CasIR were used to amplify the RPB6 GEC from its pLIB plasmid, generating a fragment with α and β overlaps. This was then assembled

with PCR-amplified pBIG1a:1 to generate a final pBIG1a:6-1 plasmid. This two-step assembly method was repeated for three more vectors containing 6 subunit GECs: RPB10 and RPB3 assembled into pBIG1b, RPB9 and RPB2zz into pBIG1c, lastly RPB7 and RPB5 into pBIG1d.

## Constructing pBIG2 vectors

For co-expression of the 8-subunit ASFV RNAP, we designed two pBIG2 vectors of pBIG2bc:RPB3-RPB10-RPB9-RPB2zz and pBIG2ad:RPB6-RPB1-RPB5-RPB7. The former construct required producing a new vector backbone, generated by amplifying the provided pBIG2abc vector, with primers DampF and GAαtoBR (Supplementary Table 3) generating the linear pBIGbc plasmid. Poly-gene expression cassettes (PGC) of 'B-RPB3-RPB10-C' and 'C-RPB2zz-RPB9-D' were digested with PmeI (NEB, R0560S). Gibson assembly was carried out with pBIGbc and the two PGC fragments at a ratio of 1:3:3. The pBIG2bc:RPB3-RPB10-RPB9-RPB2zz assembly was assessed by restriction digestion and Sanger sequencing. In contrast, the method for assembling pBIG2ad:RPB6-RPB1-RPB5-RPB7, used the biGBac system vector pBIG2ad 'as-is', following PmeI digestion. The PGC fragment A-RPB6-RPB1-B was PmeI-digested from the pBIG1a vector directly. However, the D-RPB5-RPB7-E PGC was PCR-amplified with primers GADtoBF and EampR (Supplementary Table 3) to switch the D overlap for a B overlap, producing the PGC B-RPB5-RPB7-E. These PGC fragments were assembled with PmeI-digested pBIG2ad vector in a ratio of 3:3:1 (respectively), and Sanger sequenced.

## Expression of ASFV RNAP

The two plasmids (pBIG2bc:RPB3-RPB10-RPB9-RPB2zz and pBIG2ad:RPB6-RPB1-RPB5-RPB7) were each transformed into chemically competent DH10 EMBacY *E. coli* cells (Geneva Biotech). Cells were then plated onto LB agar plates containing gentamycin, tetracycline, kanamycin, IPTG and X-gal and incubated at 37 °C overnight. Colonies were picked via blue-white selection for overnight liquid culture and then, the bacmid DNA isolated by alkali denaturation utilising buffers P1, P2 and N3 from QIAprep Spin Miniprep Kit (QIAGEN) followed by isopropanol precipitation. Sf9 or High Five cells (Gibco™, ThermoFisher Scientific, catalogue numbers 11496015 and B85502, respectively), grown in the Insect-XPRESS Insect Cell Culture Media with L-Glutamine (Lonza) supplemented with Penicillin-Streptomycin (Gibco), were transfected with each recombinant bacmid using FuGENE HD Transfection Reagent (Promega) according to the manufacturer´s protocol and incubated at 28 °C for 3 days. These cultures were used to generate baculovirus V0 stocks stored at 4 °C, and these stocks were subsequently used to simultaneously co-infect larger scale cultures of High Five cells for protein expression of the complete ASFV RNAP. Baculovirus infection progress was monitored via YFP expression using Luna-FL Dual Fluorescent Cell Counter (Logos Biosystems). Cultures were pelleted by centrifugation at 2000 x g and flash frozen using liquid nitrogen, before storing at −80 °C.

## Purification of ASFV RNAP

Cell pellets were resuspended in N250 buffer (50 mM Tris-HCl pH 8.0, 250 mM NaCl, 1 mM MgCl$_2$, 0.1 mM ZnSO$_4$, 10% glycerol and 1 mM DTT) supplemented with a proteases inhibitor (cOmplete EDTA-free Protease Inhibitor Cocktail Tablet, Roche) and 50 µg/ml DNase I. Resuspended cell pellets were incubated on ice for 1.5 h vortexing briefly every 30 mins. Cells were lysed with a Dounce homogeniser on ice (30 plunges) and cell lysate was centrifuged at 50,000 x g for 30 mins at 4 °C, then, the resulting supernatant was filtered using 1.2 µm Minisart Syringe Filter (Sartorius). ASFV RNAP was purified by affinity via the N-terminal ZZ-tag on vRPB2. In brief, IgG Sepharose 6 Fast Flow (Cytiva) beads were equilibrated by washing them twice in N250 buffer, followed by centrifugation at 1000 x g for 2 mins. The equilibrated beads were resuspended in the soluble cell lysate and incubated for 1 h on ice on a shaker. A gravity column (Poly-Prep

Chromatography Column, Bio-Rad) was first washed with N250 buffer, before the beads-sample mix was applied and the flow-through collected. This flow-through would undergo multiple rounds of re-incubation with freshly equilibrated beads until no more eluted protein could be observed via SDS-PAGE. The beads were washed with N250 one last time and finally, resuspended in 2 ml of N250 buffer. For elution of vRNAP from the IgG Sepharose beads, 20 µl of 1 mg/ml TEV protease (NEB, Cat no. P8112S) was added to each 2 ml aliquot of bead-bound protein and incubated at room temperature for 1.5 h on a shaker. Samples were then applied to a fresh N250 buffer pre-equilibrated gravity column, and the eluate collected. The cleaved sample eluted from the column was concentrated using Amicon Ultra-15 Centrifugal Filter Unit 100 kDa MWCO (Millipore) before applying to size exclusion by means of a tandem Superose 6 Increase 10/300 GL (Cytiva) and Superdex 200 Increase 10/300 GL (Cytiva) columns equilibrated in N250 buffer. Concentration and quality of eluted protein was assessed via Qubit (Invitrogen, Thermo Fisher Scientific) and SDS-PAGE with 4–20% Mini-PROTEAN TGX Precast Protein Gels (BIO-RAD). Following this protocol, we routinely obtained around 2.5 mg of recombinant ASFV RNAP per litre of insect cell expression culture.

## Mass spectrometry analysis

For mass spectrometry analysis, samples were run on SDS-PAGE, and bands were excised using a scalpel cleaned with 70% EtOH before each extraction. Bands of interest were shipped to the Mass Spectrometry and Proteomics facility (St Andrews University) to be analysed by mass spectrometry on a nanoLCMSMS instrument (Fusion Lumos, ThermoScientific) using a EasySpray 15 cm Pepmap column. The acquired peptide spectra were searched against the NCBI database of protein sequences, specifically databases 'contaminants 20160129', 'BMS 211207' and 'cRAP 20190304' using the Mascot search algorithm[77] (MatrixScience) and the best matches are listed in Supplementary Table 1.

## Nonspecific in vitro transcription assay

To assess transcription activity in the SEC fractionation we carried out nonspecific in vitro transcription assay[20]. Recombinant RNAP was incubated with 500 µM ATP/GTP/CTP (ThermoFisher Scientific), 1 µM UTP, 0.2 µl 3000 Ci/mmol [α-$^{32}$P]-UTP (Hartmann Analytic), and 350 ng of Activated Calf Thymus DNA template type XV (Sigma-Aldrich) in 50 µl transcription buffer (25 mM Tris-HCl pH 8.0, 50 mM KCl, 3 mM MgCl$_2$, 2 mM DTT, 100 µg/ml BSA, 1 U RNasin (Promega). Reactions were allowed to proceed for 30 min at 37 °C and stopped by transferring into 1 ml of ice cold 5% Trichloroacetic acid (TCA, Sigma-Aldrich) and incubated for another 15 min on ice. Precipitated nucleic acids were collected on circular 25 mm Glass Microfiber GF-F Filters (GE Healthcare), washed with ice-cold 5% TCA, and the filter was transferred into 5 ml of Ecoscint A Scintillation Cocktail (National Diagnostics) in a scintillation tube. The signal was measured as counts per minute (CPM) using a Tri-Carb 2900TR Liquid Scintillation Counter (PerkinElmer) with 1 min signal count time.

Following further optimisation, we adjusted the reaction conditions for subsequent experiments: 1 µg of ASFV RNA polymerase, 250 ng activated calf thymus DNA template, 5 mM MgCl$_2$, 0.5 mM GTP, 0.5 mM CTP, 0.5 mM ATP, 6 µM UTP, 0.1 mg/ml recombinant albumin (NEB), 1.5 mM DTT, 25 mM HEPES pH 8.0, 0.2 µl 3000 Ci/mmol [α-$^{32}$P]-UTP (Hartmann Analytic). Components were mixed on ice, then incubated for 10 minutes at 37 °C, or in a range between 10 to 50 °C. Reactions were stopped by placing on ice and addition of 60 mM EDTA-NaOH pH 8.0 before transferring to 0.8 ml cold 5% TCA. For pH range, 25 mM of either MES-NaOH pH 6.0, HEPES-NaOH pH 7.0/8.0, Tris-HCl pH 9.0 adjusted at 37 °C. For ionic strength, a range of KCl concentrations from 0 to 100 mM was used. MgCl$_2$ and MnCl$_2$ were used in the reaction mixture at a concentration range between 0.75 and 10 mM. For DNA template comparison, 250 ng of either activated

calf thymus DNA template (Sigma), or its denatured form obtained by exposure to 100 °C for 10 min followed by rapid cooling on ice, or M13mp18 single-stranded DNA (NEB), or double-stranded M13mp18 RF I DNA (NEB), all in storage buffer 10 mM Tris-HCl pH 8.0 with 1 mM EDTA. Finally, for the inhibitor screening either 100 nM *E. coli* core RNAP (NEB, in provided buffer), 100 nM RNAPII from *S. cerevisiae*, or 100 nM ASFV RNAP were supplemented with 5% DMSO in the control reaction or pre-incubated at rt for 10 min with the appropriate inhibitor in 5% DMSO: 100 μM alpha-amanitin from *Amanita Phalloides* (Apollo Scientific), 100 μM Rifampicin (LKT Laboratories Inc.), 50 mM EDTA-NaOH pH 8.0, or 100 μM Actinomycin D from *Streptomyces sp.* (Merck) pre-incubated with DNA template at room temperature.

In the nonspecific transcription assay, the specific activities of the ASFV core RNAP is 83 ( ± 8) nmol incorporated UMP per hour per mg RNAP (or 37 ± 4 nmol incorporated UMP per hour per nmol RNAP), *S. cerevisiae* RNAPII 20 ( ± 2) nmol incorporated UMP per hour per mg RNAP (or 10 ± 1 nmol incorporated UMP per hour per nmol RNAP,) and the *E. coli* core RNAP 130 ( ± 7) nmol incorporated UMP per hour per mg RNAP (or 51 ± 3 nmol incorporated UMP per hour per nmol RNAP). Source data are provided as a Source Data file.

### Cryo-EM data collection and processing
The recombinant ASFV RNA polymerase sample at 0.9 mg/ml in N250 buffer (containing 5% glycerol), was diluted up to 0.05 mg/ml in the same buffer without glycerol. Then, 3 μl of the diluted sample was spotted on a C-flat holey grid 400 mesh R1.2/1.3 (Electron Microscopy Sciences, USA), glow discharged in air, 1 min at 25 mA and 0.2 mBar (Pelco easiGlow, Ted Pella) covered with a layer of freshly prepared graphene oxide following a protocol previously described by Cheng K, et al. [78], and vitrified in liquid ethane by using a Vitrobot Mark IV (Thermo Fisher Scientific, USA) at 4 °C and 95% humidity. Data were collected in the ISMB Birkbeck EM facility using a Titan Krios D3771 microscope (Thermo Fisher Scientific, USA) operating at 300 keV, equipped with a BioQuantum energy filter and a post GIF K3 direct electron detector (Gatan, USA). The images were collected in super-resolution mode, at a nominal magnification of 105,000, applying on-the-fly a binning of 2 providing images with a final pixel size of 0.828 Å, with a dose rate of 12.7 e⁻/px/sec, and a total dose of 48.152 e⁻/Å², fractionated over 50 frames. An energy slit with a 20 eV width was used during data collection. Data were collected using EPU2 software with a nominal defocus range -1.5 μm to -2.7 μm.

A dataset of 14,638 movie stacks was aligned and summed using Relion v4[79] implementation, followed by CTF estimation using CTFFIND v4.1[80]. Topaz[81] was used for the particle picking which led to the extraction of 4,842,857 particles, initially downscaled to 2.484 Å. After a few cycles of 2D and 3D classifications in cryoSPARC v3[82], 3,825,942 good particles were selected and rescaled to the original pixel size of 0.828 Å. During 3D classification of both original size and downscaled particles, multiple intermediates, from closed to open conformations, of the DNA-binding channel were observed. In order to identify and describe the conformational heterogeneity within the dataset, after a cycle of homogeneous refinement in cryoSPARC v3, the roughly four million selected particles were analysed through 3D variability analysis[25] using three eigen vectors. The results were output both as movies of ten frames for each vector, and particle clusters to isolate the batch of particles corresponding to the closed conformation, with 467,519 particles, and the widest open conformation available in the dataset, which corresponded to 352,192 particles. The two batches of particles corresponding to the closed and open conformations were exported back into Relion v4, where, after refinement and post-processing, they were subjected to CTF refinement and particle polishing. The final cycle of 3D refinement and post-processing resulted in a map with a nominal resolution of 2.73 Å for the closed conformation and 2.92 Å for the open conformation. The nominal resolution was estimated using the gold standard Fourier

Shell Correlation (FSC) with a 0.143 threshold. To reduce the flexibility and enhance the map quality of the stalk domain, we performed a multi-body refinement using the RNAP in the closed conformation as main domain and the corresponding stalk as moving body. To do that, we generated two masks with soft edges for the large globular main domain and the stalk, respectively, to which we applied 20 ° of width for the rotation priors and 5 pixels for the translation between the two bodies, as described by Nakane T and Sheres SHW[27], as well as successfully used for the crenarchaeal RNAP[39]. The multi-body refinement improved the stalk resolution up to 2.99 Å after post-processing (Supplementary Figs. 2, 3 and Table 2). The local resolution was assessed using Relion v4 and rendered in UCSF Chimera[83]. The batch of particles corresponding to the closed conformation was further processed in cryoSPARC v4.1 using the demo version of the 3D flexible refinement (3DFlex)[29]. After a non-uniform refinement, the mesh was prepared with default parameters using a mask with generous soft edges. The 3DFlex refinement modelling was repeated several times and the final optimal parameters were: number of latent dimensions 5, rigidity (lambda) 0.5, latent centering strength 15; all the other parameters were left as default. The model was used to reconstruct a high resolution map and corresponding half maps to be used directly for FSC calculation (nominal resolution 2.69 Å), and local resolution evaluation, while map sharpening was carried out with Autosharpen in Phenix v1.20[84]. Because 3D flex is a demo version, we reported in the Supplementary Information the validation results for both standard 3D and 3DFlex refinements for comparison (Supplementary Figs. 3, 4 and Table 2).

### Model building and refinement
Map sharpening was carried out with Autosharpen in Phenix v1.20 applying overall anisotropy removal for both the closed and open conformations, and stalk maps. Then, using the Combine Focused Maps programme (Phenix v1.20), the stalk map was combined with the sharpened closed and open conformation maps to generate the complete maps. Models of all RNAP subunits were generated with AlphaFold2[28] via AlphaFold Colab (https://colab.research.google.com/github/deepmind/alphafold/blob/main/notebooks/AlphaFold.ipynb) on Google Colab. The full RNAP model was assembled using the homologous human RNAPII as reference model in Chimera. After centring the model inside the cryo-EM map of the closed conformation, model refinement was carried out in Phenix v1.20, alternating it with rounds of manual editing and refinement in Coot v0.9.8.7[85] to correct the initial AlphaFold2 models, following the map density, and place magnesium and zinc ions. To overcome the displacement of the side chains of arginine and lysine residues surrounding the two extra-densities present in the map (despite the side chains densities being visible, the closest chains were dragged inside the larger extra-densities), one molecule of CTP was added in each density to rebalance the refinement requirement. The addition was sufficient to ensure correct refinement of the side chains, while the two molecules of CTP were subsequentially removed at the end of the refinement. The structure obtained was used to refine the same model against the map of the open conformation. Finally, for the closed conformation, the map obtained from 3DFlex was used to address the flexibility of the protrusion and wall domains. The map allowed the introduction of roughly 100 residues that were initially excluded from the refinement because of insufficient map density to prove the correct folding. Interestingly, the Combine Focused Maps tool was not necessary because the stalk domain was obtained at similar resolution and map quality compared to the multibody refinement map. The refinements of the two conformations were evaluated and validated using Mol-probity webserver[86] (Table 2).

### Structure-based sequence alignments
Structure-based alignments were carried out using the MatchMaker tool in Chimera. Each subunit of the ASFV RNAP was superimposed,

using the BLOSUM-62 matrix and a RMSD cutoff of 10 Å as parameters, with the equivalent subunit from *H. sapiens* (pdb 5iyc[42]), *S. cerevisiae* (pdb 7o75[87]), *S. acidocaldarius* (pdb 7ok0[39], and 7oqy[39] for Rpo7 only), *T. kodakarensis* (pdb 6kf3[88]), and VACV (pdb 7amv[15]), and then, the alignment generated was manually edited to account for the higher flexibility of loops and disordered terminal tails. VACV RPB11 was removed from the alignment because the superimposition failed to provide a reasonable result against both ASFV and human RPB11, suggesting that the low sequence similarity of VACV RPB11 with all the other species is reflected by a poor structural conservation. All alignments were rendered using Esprit 3[89], using the percentage equivalent as parameter with a global score above 0.6.

For the CTD of ASFV RPB7 and human 2´O-MTase 1, the Match-Maker tool of Chimera did not work given the low structure conservation between the two structures (Z-score from DALI[43] search was only 6.3). To overcome the issue, the DALI structural equivalences information was used to improve the superposition which was obtained by using a BLOSUM-35 matrix (instead of the default BLOSUM-62 matrix used in Chimera). The structure-based sequence alignment obtained from the superposition was implemented with the refined vRPB7 structure of the closed conformation and the crystal structure of the human 2´O-MTase 1 (pdb 4n48[44]) to visualise the secondary structure elements using Esprit 3 rendering and default labelling.

### Reporting summary

Further information on research design is available in the Nature Portfolio Reporting Summary linked to this article.

## Data availability

Source data are provided with this paper as Source Data file. The data generated during the current study are available in the Protein Data Bank with the following PDB codes: 8Q3B for the closed conformation, and 8Q3K for the open conformation. The corresponding cryo-EM maps have been deposited on the Electron Microscopy Data Bank, with the following codes: EMD-18120 for the closed conformation and EMD-18129 for the composite map of the open conformation. In addition, we deposited the maps used to generate the composite map for the open conformation with the codes EMD-18163 for the consensus map, and EMD-18164 for the map of the stalk domain obtained from multibody refinement. The raw cryo-EM data are available upon request by writing to the corresponding author. The structures used for this study are: 7D8U (ASFV N7-MTase domain), 4N48 (human 2´O-MTase), 6RIE (VACV CCC), 3K1F (yeast RNAPII/TFIIB), 6RFL (VACV inactive PIC), 5IYC (human PIC-OC, open complex), 7O75 (yeast RNAPII PIC-OC), 7OK0 (*S. acidocaldarius* RNAP), 7OQY (*S. acidocaldarius* RNAP/TFS4), 6KF3 (*T. kodakarensis* RNAP) and 7AMV (VACV PIC-OC). Source data are provided with this paper.

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

## Acknowledgements

African Swine Fever Virus research in the RNAP laboratory at UCL is funded by BBSRC grant BB/X017028/1 awarded to FW. Further work is funded by Wellcome Trust Investigator in Science Award WT207446/Z/17/Z awarded to FW. Mass spectrometry analysis was carried out at St Andrews University Mass Spectrometry and Proteomics facility (BBSRC grant BB/T027686/1). Cryo-EM data for this investigation were collected at ISMB EM facility (Birkbeck College, University of London) with financial support from the Wellcome Trust (202679/Z/16/Z and 206166/Z/17/Z). We would like to acknowledge Dr Jerome Gouge at the ISMB for his assistance with insect cell expression and Dr Anthony Roberts for supplying the baculovirus expression vectors used in this study. We thank Dr Alan Cheung for providing yeast RNAPII, and earlier members of the RNAP laboratory Drs Carol Sheppard and Thomas Fouqueau for their help during the early stages of the project. We are indebted to Drs Linda Dixon and Chris Netherton at the BBSRC Pirbright Institute for their invaluable advice on ASFV biology, and Dr Utz Fischer at the University of Würzburg in Germany for inspired discussions about NCLDV transcription. Finally, we'd like to thank Dr Tine Arnvig at the ISMB for critical reading of this manuscript. We like to dedicate this work to the late Dr Pierre Thuriaux at the CEA Saclay in Gif-sur-Yvette, France. Pierre predicted the compelling similarity of ASFV and archaeal RNAPs based on very limited sequence information more than 25 years ago and served as a great inspiration for this project.

## Author contributions

G.C. and F.W. conceived the original idea and devised the project. G.C. and M.S. contributed to the sample cloning, expression and purification; M.S., and C.D. performed the enzymatic assays' experiments, S.P. carried out the structural studies by cryo-EM. Manuscript conceptualisation, F.W.; writing, F.W., S.P., G.C., M.S and C.D.; figure preparation, S.P., M.S., G.C., and C.D. All authors have read and agreed to the published version of the manuscript.

## Competing interests

The authors declare no competing interests.
