## [Peer Review File · Nature Communications]

Structure of the recombinant RNA polymerase from African Swine Fever VirusREVIEWER COMMENTS

Reviewer #1 (Remarks to the Author):

African swine fever virus (ASFV) belongs to the family of Asfarviridae within the group of NLCDDVs. Closely related to poxviruses, ASFV causes a hemorrhagic fever linked to high mortality in domestic pigs with a recent, dramatic economic impact on the pork industry in Asia and Europe. Due to this reason, ASFV is considered a potential candidate to be released in an act of bioterrorism and a corresponding preparedness is desirable.

Like all NLCDDVs, ASFV possesses a multi-subunit RNAP active in the host cell cytoplasm with indubitable homology to its well-characterized poxviral counterpart that is an attractive target for rational design of anti-ASFV drugs.

Here, Pilotto et al. from the Werner lab have solved the non-trivial task to establish a reconstitution procedure for ASFV RNAP expressed from two different polycistronic baculovirus strains. This is a prerequisite to deal with the ASFV transcription machinery within a normal lab infrastructure as handling of live ASFV itself requires the highest biosafety standards. Their heterologously expressed vRNAP preparation is active in a non-promoter specific transcription assay and they present the 2.7/2.9A cryo EM structures of the apo ASFV RNAP in the open and closed cleft states.

As expected from sequence homology, ASFV RNAP displays significant structural similarity to Pol II and vaccinia RNAP. Still, the authors identify several important differences, including the structural cause of insensitivity to amanitin and the integration of a methyltransferase domain into the stalk. Resemblance to the methyltransferase of the vaccinia capping enzyme allows for modelling of a putative ASFV co-transcriptional capping complex (CCC).

The cryo EM structure determination is technically sound, and the manuscript is well written. The presented structures are of scientific significance and hence the manuscript deserves publication after the following points have been addressed:

Major points:

- The high structural similarity of the catalytic core to that of Pol II and vaccinia RNAP should enable superposition and/or modelling of an actively transcribing complex. A corresponding supplemental figure would be helpful.
- Would the ribbon domain of TFII B/Rap94 fit into ASFV RNAP?
- Please give some background information regarding the ASFV capping system, are the TPase and GTase functions located in separate proteins?
- Since the vaccinia capping system serves as the blueprint for the CCC model shown in Fig. 6 d), please show it in comparison (the vaccinia CCC structure allows for identification of the full RNA path). Please mark also the active sites and the RNA exit pore of the RNAP.

Minor points:

Figure 3/4: The font size of the domain names is too small and the contrast against the colored background too weak. The labels should be placed outside of the domain scheme. Please label also the cartoon depictions.

Figure 5 appears to be missing in the manuscript. The corresponding reference in the text seems to refer to Fig. 6 (re-labelling of Fig. 6 to Fig. 5 seems to solve the problem).

Tab. 1: Rpo30 (homologous to TFIIS) is a stably integrated subunit in vRNAP, please add.

Supplementary Figure 5: Legends to d) and e) are missing.

P3 L88: The comma should be removed.

P7 L181 „a high structural similarity“ instead of „a good structural similarity“

P7 L183 „accurately “ instead of „accurate“

P17 L387 Please cite reference for RNAP genes being transferred back and forth from viruses to eukaryotes.

Reviewer #2 (Remarks to the Author):

Prof. Werner and colleagues provide an important advance in understanding the transcriptional apparatus of ASFV, a large cytoplasmic DNA virus that threatens the pork industry.

In a series of technically adroit experiments, they produced a recombinant version of ASFV RNAP by co-expression of the 8 known/predicted subunits in insect cells, used an affinity tagged subunit to attain initial purification (followed by tag cleavage and gel filtration), and performed biochemical assays to validate “non-specific” RNA synthesis activity. They go on to solve cryo-EM structures of the RNAP at 2.7–2.9 Å resolution, thereby revealing features conserved with vaccinia virus (VACV) RNAP and cellular Pol2, plus structural elements unique to the ASFV RNAP. Among the latter is a domain of vRPB7 that is plausibly suggested to be a docking site on RNAP for the ASFV cap guanine-N7 methyltransferase.

The work is of high interest with respect to evolution and diversity of the eukaryal transcriptional machinery and has ramifications for enabling anti-ASFV drug discovery (an urgent need absent a veterinary vaccine for ASFV).

Technical clarifications:

1) Whereas the assay for non-specific RNA polymerase activity is described in detail in Methods, the quantification of the extent of UMP incorporation is unclear. The y-axis in Fig. 1d is denoted as % of UMP incorporation by each fraction (presumably of the total of all fractions). Whereas this suffices to establish an elution profile, it does little to convey how much RNA is synthesized. Things are expressed more quantitatively in the optimization assays in Fig. S1 as pmol UMP incorporated per hour per µg of protein. Yet the optimum values of 0.04 seem quite low. Taking the reported MW of 400 kDa for RNAP, this would correspond to a yield of 40 fmol of UMP per 2.5 pmol of RNAP in 1 hour (unless I’m mistaken here). The authors should report the specific activity of the enzyme, as they calculate it, and comment on the low value.

2) A potential caveat about the polymerase assay is that “non-specific” polymerase activity, e.g., for vaccinia RNAP, is optimally assayed using heat-denatured (i.e., single-stranded) DNA as template and

manganese as the divalent cation cofactor. In this regime, the VACV RNAP is extremely active (see Spencer et al. JBC 1980). It would be nice to see activity of the ASFV enzyme measured under similar conditions, i.e., it is most unlikely that the so-called “optimal” assay conditions described here are the best way to measure non-specific RNA synthesis by the ASFV enzyme. This will be important for any assays aimed at screening for inhibitors. I encourage the authors to measure and report activity with heat-denatured DNA and Mn in this study.

Some text issues requiring revision:

- 1) Abstract, line 23: delete the phrase “offering grounds for the development of highly selective inhibitors.” The logic of the sentence is flawed – the fact that the ASFV Pol is resistant to rifampin and amanitin has no probative value regarding discovery of specific inhibitors. The final sentence in the Abstract suffices on this point.
- 2) Intro, line 63: it is simply not true that “all VACV experiments are reliant on affinity-purified RNAP complexes isolate from HeLa cells infected with VACV.” Rather, most of what is known regarding the functional properties of VACV RNAP, including its subunit composition and the discovery (via reconstitution biochemistry) of the associated factors required for early transcription initiation and termination was discovered using native RNAP isolated from purified infectious virions (without the benefit of affinity tags). The authors should delete (or modify) this claim.
- 3) Line 143: “scintillation counting”
- 4) There is no legend for panels d and e in Supplementary Figure S5.
- 5) Line 200: I can’t see a basis for the claim re the “extra densities” that “none of the two ligands seem to affect RNAP activity.” Ditto “flexibility”. How do the authors know this? Perhaps activity would be much higher if the densities were not there. This claim should be deleted. It suffices to say “we cannot assess their physiological significance.”
- 6) Page 15: “Figure 6” – shouldn’t it be Figure 5?
- 7) Discussion, line 360: the authors state they purify ASFV RNAP at “suitable milligram scale” but I don’t see any direct information in the Methods section (p. 22) on the yield of RNAP after the gel filtration step. If they are going to make this claim, then that information (yield of RNAP per X amount of insect cells) must be provided.

Point to point response to reviewers' concerns

(our responses are highlighted in blue, new MS text in italics)

Reviewer #1 (Remarks to the Author):

African swine fever virus (ASFV) belongs to the family of Asfarviridae within the group of NLCDDVs. Closely related to poxviruses, ASFV causes a haemorrhagic fever linked to high mortality in domestic pigs with a recent, dramatic economic impact on the pork industry in Asia and Europe. Due to this reason, ASFV is considered a potential candidate to be released in an act of bioterrorism and a corresponding preparedness is desirable.

Like all NLCDDVs, ASFV possesses a multi-subunit RNAP active in the host cell cytoplasm with indubitable homology to its well-characterized poxviral counterpart that is an attractive target for rational design of anti-ASFV drugs.

Here, Pilotto et al. from the Werner lab have solved the non-trivial task to establish a reconstitution procedure for ASFV RNAP expressed from two different polycistronic baculovirus strains. This is a prerequisite to deal with the ASFV transcription machinery within a normal lab infrastructure as handling of live ASFV itself requires the highest biosafety standards. Their heterologously expressed vRNAP preparation is active in a non-promoter specific transcription assay and they present the 2.7/2.9A cryo EM structures of the apo ASFV RNAP in the open and closed cleft states.

As expected from sequence homology, ASFV RNAP displays significant structural similarity to Pol II and vaccinia RNAP. Still, the authors identify several important differences, including the structural cause of insensitivity to amanitin and the integration of a methyltransferase domain into the stalk. Resemblance to the methyltransferase of the vaccinia capping enzyme allows for modelling of a putative ASFV co-transcriptional capping complex (CCC).

The cryo EM structure determination is technically sound, and the manuscript is well written. The presented structures are of scientific significance and hence the manuscript deserves publication after the following points have been addressed

Major points:

- The high structural similarity of the catalytic core to that of Pol II and vaccinia RNAP should enable superposition and/or modelling of an actively transcribing complex. A corresponding supplemental figure would be helpful.

Following the advice of reviewer #1 we have modelled the ASFV transcription elongation complex (TEC) using RNAPII and VACV RNAP as references, and we have prepared the Supplementary Figure 10 highlighting the most interesting differences within the active site of ASFV RNAP compared to RNAPII and VACV RNAP.

To discuss the additional data, we have introduced the following sentences into the results section of the manuscript from line 426:

'Superimposition of the initially transcribing complexes of the human RNAPII or VACV RNAP with the ASFV RNAP structure allowed a closer scrutiny of the active site environment and revealed small differences in the fork loop 2 and helix (residue 794 - 801) motifs in vRPB2 (Supplementary Fig. 10). The fork loop 2 in ASFV is replaced by the less flexible alpha-helices connected by a short loop; however, the position of the loop is perfectly overlapping with the fork loop 2 in RNAPII. By contrast, the vRPB2 helix 794 - 801 runs perpendicularly towards the RNA binding site, instead of forming a stable interaction with the rest of the hybrid binding domain as seen in VACV and human RNAPII. Yet, none of the two motifs seem to interfere with the scaffold modelling.'

- Would the ribbon domain of TFIIIB/Rap94 fit into ASFV RNAP?

We have fitted the zinc ribbon and part of the linker region of human TFIIIB and VACV Rap94 into the ASFV RNAP structure. As predicted, due to the high degree of structural conservation, both fit very well into the dock domain and active site cleft of the ASFV RNAP (Figure 6b). To highlight the similarities and differences among factors, we have introduced a schematic of the domain composition of ASFV and human TFIIIB factors and VACV Rap94 in figure 6a.

To discuss the additional analyses, we have introduced the following sentences into the discussion section of the manuscript, from line 566:

'The high degree of structural conservation allowed the modelling of the B-ribbon domain of the universally conserved initiation factor TFIIIB and the DNA/RNA scaffold on the ASFV RNAP using both RNAPII and VACV RNAP structures as templates (Fig. 6). Factors related to TFIIIB are essential for transcription initiation of the archaeal RNAP (TFB), eukaryotic RNAPI (TAF1B), II (TFIIIB) and III (Brf1) as well as VACV RNAP (Rap94). Specifically, interactions between the B-ribbon of TFIIIB-like factors and the RNAP dock domain are instrumental for RNAP recruitment to promoters. The TFIIIB homolog of ASFV is encoded in the viral core genome and thought to facilitate late, and possibly intermediate, gene transcription. Both human TFIIIB and VACV Rap94 B-ribbon and B-reader domains fit well in the ASFV RNAP structure suggesting that the molecular mechanisms of TFIIIB/RNAP recognition during initiation is conserved in ASFV.'

- Please give some background information regarding the ASFV capping system, are the TPase and GTase functions located in separate proteins?

To clarify this point, we have introduced the following sentence into the discussion section lines 623-625, supported by the new Supplementary Fig. 12, panel a, where we show the domain organization of the CE in ASFV and VACV, line 619:

'The ASFV CE is homologous to the VACV D1 subunit, while ASFV does not encode obvious homologues of the VACV D12 subunit¹² (Supplementary Fig. 12a)'

- Since the vaccinia capping system serves as the blueprint for the CCC model shown in Fig. 5 d), please show it in comparison (the vaccinia CCC structure allows for identification of the full RNA path). Please mark also the active sites and the RNA exit pore of the RNAP.

We have prepared a new Supplementary Figure 12 that compares the ASFV RNAP structure with the VACV CE (CCC) structure after superposition of the ASFV and VACV RNAPs. As the VACV CE interacts with the RNAP in different ways dependent on the transcription cycle step, we have also included a panel showing the ASFV RNAP and VACV CE in the topology of the VACV transcription preinitiation complex (PIC). The superimpositions reveal clashes between the ASFV RNAP and the OB-fold domain of the VACV CE. It is important to point out that the ASFV D12 equivalent, the vRPB7-CTD, is in a different position- at the C-terminus of vRPB7. Based on the predicted interaction of vRPB7-CTD and the ASFV CE N7-MTase domain, this places the latter on the opposite side of the vRPB7 stalk in our model in figure 5d. Despite the sequence similarity of ASFV and VACV CE, in VACV the TPase, GTase, and OB-fold domains form a stable unit around which the N7-MTase moves thanks to a long 18 aa linker. In comparison, according to AF2 modelling (used to define the domain organization shown in panel a), the ASFV TPase and GTase share minimal interface contacts, the linker connecting the OB-fold to the N7-MTase is much shorter (10 aa), supporting the two domains in forming a wide interface. Overall, this suggest that neither the VACV CCC nor PIC complexes are high confidence reference models for the fitting of the ASFV CE.

To discuss the additional analyses, we have introduced the following sentences into the body text of the manuscript, lines from 629:

'While the domain organisation of the VACV CE D1 subunit and ASFV CE is conserved (TPase, GTase, OB-fold, and N7-MTase), the ASFV RNAP/CE model differs from VACV co-transcriptional capping complex (CCC) because the ASFV CE N7-MTase domain is predicted to be on the other side of vRBP7 due to the hypothetical interaction with the vRPB7 CTD (Supplementary Fig. 12b, and c). However, we note that the topology of the RNAP/CE complex is highly variable in VACV, as the CE in the context of the VACV preinitiation complex (PIC) remains proximal to the RNAP stalk but in a different orientation and displaced from the RNAP core due to the binding of other virus-specific transcription factors (Supplementary Fig. 12d).'

Minor points:

Figure 3/4: The font size of the domain names is too small and the contrast against the coloured background too weak. The labels should be placed outside of the domain scheme. Please label also the cartoon depictions.

Corrected.

Figure 5 appears to be missing in the manuscript. The corresponding reference in the text seems to refer to Fig. 6 (re-labelling of Fig. 6 to Fig. 5 seems to solve the problem).

Corrected.

Tab. 1: Rpo30 (homologous to TFIIS) is a stably integrated subunit in vRNAP, please add.

Following the reviewer's advice, we have included Rpo30 in table 1. However, we do note that Rpo30 association with VACV RNAP is context dependent and would strictly speaking not define Rpo30 as RNAP subunit.

Supplementary Figure 5: Legends to d) and e) are missing.

Added.

P3 L88: The comma should be removed.

Deleted.

P7 L181 „a high structural similarity“ instead of „a good structural similarity“

Corrected.

P7 L183 „accurately“ instead of „accurate“

Corrected.

P17 L387 Please cite reference for RNAP genes being transferred back and forth from viruses to eukaryotes.

Reference added.

Reviewer #2 (Remarks to the Author):

Prof. Werner and colleagues provide an important advance in understanding the transcriptional apparatus of ASFV, a large cytoplasmic DNA virus that threatens the pork industry.

In a series of technically adroit experiments, they produced a recombinant version of ASFV RNAP by co-expression of the 8 known/predicted subunits in insect cells, used an affinity tagged subunit to attain initial purification (followed by tag cleavage and gel filtration), and performed biochemical assays to validate “non-specific” RNA synthesis activity. They go on to solve cryo-EM structures of the RNAP at 2.7–2.9 Å resolution, thereby revealing features conserved with vaccinia virus (VACV) RNAP and cellular Pol2, plus structural elements unique to the ASFV RNAP. Among the latter is a domain of vRPB7 that is plausibly suggested to be a docking site on RNAP for the ASFV cap guanine-N7 methyltransferase.

The work is of high interest with respect to evolution and diversity of the eukaryal transcriptional machinery and has ramifications for enabling anti-ASFV drug discovery (an urgent need absent a veterinary vaccine for ASFV).

Technical clarifications:

1) Whereas the assay for non-specific RNA polymerase activity is described in detail in Methods, the quantification of the extent of UMP incorporation is unclear. The y-axis in Fig. 1d is denoted as % of UMP incorporation by each fraction (presumably of the total of all fractions). Whereas this suffices to establish an elution profile, it does little to convey how much RNA is synthesized.

In Fig. 1d, we use the nonspecific assay to assess which fractions contain catalytically active RNAP assemblies, not to convey how much RNA is produced. To improve the clarity, we have altered fig 1d to show incorporated alpha-³²P-UTP as cpm for each fraction (expressed as SEC elution volume).

Things are expressed more quantitatively in the optimization assays in Fig. S1 as pmol UMP incorporated per hour per µg of protein. Yet the optimum values of 0.04 seem quite low. Taking the reported MW of 400 kDa for RNAP, this would correspond to a yield of 40 fmol of UMP per 2.5 pmol of RNAP in 1 hour (unless I’m mistaken here). The authors should report the specific activity of the enzyme, as they calculate it, and comment on the low value.

The calculation of the specific activity of the recombinant ASFV core RNAP is important to enable a comparison with other RNAPs including the VACV RNAP. Having said that, the source of the RNAP, variations in its purity and the exact *in vitro* transcription assay conditions that vary between different labs and publications, make a direct comparison based on the literature problematic.

Following reviewer-2 considerations and recommendations, we have carried out additional assay optimisation experiments to further characterise the enzymatic activity of the ASFV core RNAP and repeated all transcription assays included in figure 1e and supplementary Fig. 1; a detailed description of the assay conditions is included in the methods section of the manuscript. Based on the improved assay conditions, we calculated the specific activity of ASFV RNAP by converting the cpm values of alpha-³²P-UMP incorporated into TCA-insoluble material into Bq, and adjusting the values to account for the specific activity of the alpha-³²P-UTP radioisotope in the reaction (including the concentration of cold UTP and decay/age of ³²P). Using the improved assay conditions, the ASFV core RNAP has a specific activity of 83 (± 8) nmol UMP incorporation per hour at 37°C per mg RNAP. To the best of our knowledge, this value is not worryingly low. To allow a direct comparison, we measured and calculated the specific activities of the yeast RNAPII and *E. coli* RNAP preparations used in our controls (Figure 1e of the manuscript), which had a specific activity of 20 ± 2 and 130 ± 7 nmol UMP incorporation per hour at 37°C per mg RNAP, respectively. This demonstrates that the performance of the recombinant ASFV core RNAP is on a par with other cellular multisubunit DPBB RNAP.

Following reviewer-2's recommendation, we have introduced the following sentence into the revised manuscript line 216:

'In conclusion, we decided to use assay conditions for the ASFV RNAP at 37°C, pH 8.0, 5 mM MgCl₂, and using the more physiologically relevant double stranded DNA. Under these conditions, the specific activity of the ASFV core RNAP is 83 (± 8) nmol incorporated UMP per hour per mg of enzyme. This is comparable to Saccharomyces cerevisiae RNAPII, 20 (± 2) nmol h⁻¹ mg⁻¹, and Escherichia coli RNAP, 130 (± 7) nmol h⁻¹ mg⁻¹, obtained using our assay conditions, and also in agreement with results from the literature including the archaeal Methanocaldococcus jannaschii RNAP, whose activity, 160 nmol h⁻¹ mg⁻¹, was measured under similar assay conditions.'

2) A potential caveat about the polymerase assay is that “non-specific” polymerase activity, e.g., for vaccinia RNAP, is optimally assayed using heat-denatured (i.e., single-stranded) DNA as template and manganese as the divalent cation cofactor. In this regime, the VACV RNAP is extremely active (see Spencer et al. JBC 1980). It would be nice to see activity of the ASFV enzyme measured under similar conditions, i.e., it is most unlikely that the so-called “optimal” assay conditions described here are the best way to measure non-specific RNA synthesis by the ASFV enzyme. This will be important for any assays aimed at screening for inhibitors. I encourage the authors to measure and report activity with heat-denatured DNA and Mn in this study.

We completely agree that the activity in nonspecific transcription assay varies with the assay conditions, including the DNA template used, divalent cations (Manganese or Magnesium), and reaction buffer and substrate concentrations.

We use double-stranded DNA as template because it is *more physiologically relevant* compared to ssDNA, and magnesium as divalent cation in the reaction buffer. Manganese can substitute for magnesium in many RNAPs, but due its ‘softer’ chelation properties, manganese decreases the fidelity of NTP incorporation, and is more ‘forgiving’ for suboptimal performance of RNAP (e. g. of active site mutants of RNAPs or DNAPs, e. g. <https://pubmed.ncbi.nlm.nih.gov/36797604/>). We do agree with the reviewer that the comparison of our reaction conditions with results published in literature (including Spencer et al. JBC 1980) is helpful to establish the best assay conditions for inhibitors screening. Accordingly, we have carried out additional nonspecific *in vitro* transcription assays using equal amounts of heat-denatured calf thymus DNA, non-heat denatured calf thymus DNA, and using single- (ssDNA) and double-stranded (dsDNA) stranded M13 DNA. Finally, we tested a range of manganese and magnesium concentrations. In brief, ssDNA resulted in a 4-fold higher activity than dsDNA, similar to VACV (and other) RNAP, while using manganese (at 5 mM) gave a ~3.5-fold lower activity compared to magnesium (at 5 mM). Prior heat treatment of the calf thymus DNA increased the activity about 2-fold, but this step introduced an additional element of inaccuracy due to the heterogenous nature of the activated calf thymus DNA. All new results are included in Supplementary Figure 1.

We have included the additional results in our manuscript line 204:

'To further characterize the ASFV RNA polymerase we tested a range of pH values, ionic strengths, temperatures, divalent cations, and different DNA templates. The results showed that the ASFV RNAP has an optimum at a temperature between 30 and 40°C and pH 8.0 (Supplementary Fig. 1a-b). It is interesting to point out that the virus replicates in two very different hosts, in pigs which have a body temperature of 39°C, and in soft ticks whose body temperature fluctuates with the environment. ASFV RNAP is sensitive to ionic strength in the nonspecific assay, it is most active using low KCl concentrations (Supplementary Fig. 1c) and has a clear preference for magnesium over manganese (Supplementary Fig. 1d and 1e, respectively). The results obtained with the recombinant core RNAP expressed in insect cells are in good agreement with ASFV RNAP preparations isolated from virions. Like other RNAP, the activity of ASFV RNAP in a nonspecific assay is higher with single stranded compared to double-stranded DNA

(Supplementary Fig. 1f). In conclusion, we decided to use assay conditions for the ASFV RNAP at 37°C, pH 8.0, 5 mM MgCl₂, and using the more physiologically relevant double stranded DNA.'

Some text issues requiring revision:

1) Abstract, line 23: delete the phrase “offering grounds for the development of highly selective inhibitors.” The logic of the sentence is flawed – the fact that the ASFV Pol is resistant to rifampin and amanitin has no probative value regarding discovery of specific inhibitors. The final sentence in the Abstract suffices on this point.

We agree and have removed this statement.

2) Intro, line 63: it is simply not true that “all VACV experiments are reliant on affinity-purified RNAP complexes isolate from HeLa cells infected with VACV.” Rather, most of what is known regarding the functional properties of VACV RNAP, including its subunit composition and the discovery (via reconstitution biochemistry) of the associated factors required for early transcription initiation and termination was discovered using native RNAP isolated from purified infectious virions (without the benefit of affinity tags). The authors should delete (or modify) this claim.

This is a misunderstanding, we only refer to the source materials for the structure determination of VACV RNAP complexes, not the vast number of experiments using biochemistry, cell biology and genetics approaches. To clarify this point, we have modified the sentence, now line 80:

'However, all structural analyses of VACV RNAP are currently reliant on affinity purified RNAP complexes isolated from HeLaS3 cells infected with VACV, which yield a diverse range of RNAP complexes that represent different stages of transcription complex assembly.'

3) Line 143: “scintillation counting”

Corrected.

4) There is no legend for panels d and e in Supplementary Figure S5.

We included legends for panels d and e in the Supplementary Information file:

*'**d-e**) The two extra-densities are shown next to the bridge helix in the DNA-binding channel (**d**) and bound to the foot domain of RPB1 (**e**). All residues involved in the binding of these two unidentified ligands are labelled.'*

5) Line 200: I can't see a basis for the claim re the “extra densities” that “none of the two ligands seem to affect RNAP activity.” Ditto “flexibility”. How do the authors know this? Perhaps activity would be much higher if the densities were not there. This claim should be deleted. It suffices to say “we cannot assess their physiological significance.”

We agree with the reviewer and have changed the sentence line 302:

'Whether the ligands are physiologically relevant and affect RNAP activity or its conformational flexibility remains unknown.'

6) Page 15: “Figure 6” – shouldn't it be Figure 5?

Corrected.

7) Discussion, line 360: the authors state they purify ASFV RNAP at “suitable milligram scale” but I don't see any direct information in the Methods section (p. 22) on the yield of RNAP after the gel filtration

step. If they are going to make this claim, then that information (yield of RNAP per X amount of insect cells) must be provided.

We have introduced the yield of recombinant ASFV RNAP expressed in insect cells in the results section, line 150:

'The expression and purification method employed delivered a pure sample with a reproducible yield of typically 2.5 mg of core RNAP per litre of insect cell culture.'

And in the methods section, lines 942:

'Following this protocol, we routinely obtained around 2.5 mg of recombinant ASFV RNAP per litre of insect cell expression culture.'

REVIEWERS' COMMENTS

Reviewer #1 (Remarks to the Author):

The authors have now submitted a revised manuscript that sufficiently addresses all concerns raised by the reviewers. In particular, they have provided a comprehensive comparison to the closely related Pol II which is now illustrated in Fig. 6a and enhanced the discussion in this regard.

The manuscript should thus now put forward for publication.

Reviewer #2 (Remarks to the Author):

The authors have addressed most of the issues raised in the initial referee reports.

Several remaining issues with the manuscript text content are as follows:

1) The claim on line 53 regarding “fully transcription competent extracts from virus particles” cites the 1989 study by Caeiro and Costa (incorrectly, I think), insofar as that paper reports on transcription activity of a cytoplasmic extract of ASFV-infected cells – not on isolated virions. Whereas Caeiro and Costa show that the observed transcription activity was insensitive to amanitin, their data do not show independence of host factors (line 52). If the authors can cite a paper showing mRNA synthesis by isolated ASFV virions, then such citation should be included. For example, Salas Kuznar and Vinuela (1981) *Virology* PMID: 6168100 might fit the bill.

2) Lines 312-314 referring to recent work showing that targeting of ASFV transcription is a promising therapeutic drug discovery platform cites references 62-64, none of which has anything to do with ASFV. The titles indicate these papers are about RPB8. It is not clear what papers in the reference list actually make the point.

3) Line 312 cites reference 61 anent classical swine fever, but the cited paper is about evolutionary genomics. What is the correct reference?

4) Line 387, refers to VACV D12 acting as an allosteric activator for N7 methylation by D1, yet cites reference 2, which has nothing to do with that topic. The correct references for that statement are: Mao & Shuman (1994) *J Biol Chem* PMID: 7929111; and Schwer et al. (2006) *J Biol Chem* PMID: 16707499. Please add these two references.

In sum, many of the references are wrong and the whole manuscript needs to be carefully scrutinized to make sure all citations in the text are accurate with respect to the numbered reference list.

Once these points are resolved, the revised paper should be acceptable for publication in Nat Comm

Point-to-point response

Our response in blue.

Reviewer #1 (Remarks to the Author):

The authors have now submitted a revised manuscript that sufficiently addresses all concerns raised by the reviewers. In particular, they have provided a comprehensive comparison to the closely related Pol II which is now illustrated in Fig. 6a and enhanced the discussion in this regard.

The manuscript should thus now be put forward for publication.

Reviewer #2 (Remarks to the Author):

The authors have addressed most of the issues raised in the initial referee reports.

Several remaining issues with the manuscript text content are as follows:

Reviewer-2' remaining concerns solely pertain references. We have been plagued by difficulties with our EndNote reference library. Fortunately, reviewer-2 picked up on this mistake, and we have now corrected all referencing errors throughout the revised manuscript.

1) The claim on line 53 regarding "fully transcription competent extracts from virus particles" cites the 1989 study by Caeiro and Costa (incorrectly, I think), insofar as that paper reports on transcription activity of a cytoplasmic extract of ASFV-infected cells – not on isolated virions. Whereas Caeiro and Costa show that the observed transcription activity was insensitive to amanitin, their data do not show independence of host factors (line 52). If the authors can cite a paper showing mRNA synthesis by isolated ASFV virions, then such citation should be included. For example, Salas Kuznar and Vinuela (1981) *Virology* PMID: 6168100 might fit the bill.

We have introduced the correct reference: 'Polyadenylation, methylation, and capping of the RNA synthesized in vitro by African swine fever virus.', Salas ML, Kuznar J, Viñuela E. *Virology*. 1981 Sep;113(2):484-91.

2) Lines 312-314 referring to recent work showing that targeting of ASFV transcription is a promising therapeutic drug discovery platform cites references 62-64, none of which has anything to do with ASFV. The titles indicate these papers are about RPB8. It is not clear what papers in the reference list actually make the point.

We have introduced the correct reference: 'ASFV transcription reporter screening system identifies ailanthone as a broad antiviral compound.', Zhang Y, Zhang Z, Zhang F, Zhang J, Jiao J, Hou M, Qian N, Zhao D, Zheng X, Tan X. *Virology*. 2023 Jun;38(3):459-469.

3) Line 312 cites reference 61 anent classical swine fever, but the cited paper is about evolutionary genomics. What is the correct reference?

We have introduced the correct reference: 'The potential of antiviral agents to control classical swine fever: a modelling study.' Backer JA, Vrancken R, Neyts J, Goris N. *Antiviral Res.* 2013 Sep;99(3):245-50.

4) Line 387, refers to VACV D12 acting as an allosteric activator for N7 methylation by D1, yet cites reference 2, which has nothing to do with that topic. The correct references for that statement are: Mao & Shuman (1994) *J Biol Chem* PMID: 7929111; and Schwer et al. (2006) *J Biol Chem* PMID: 16707499. Please add these two references.

We have introduced these two correct references: 'Poxvirus mRNA cap methyltransferase. Bypass of the requirement for the stimulatory subunit by mutations in the catalytic subunit and evidence for intersubunit allostery.' Schwer B, Hausmann S, Schneider S, Shuman S. *J Biol Chem.* 2006 Jul 14;281(28):18953-60., and 'Intrinsic RNA (guanine-7) methyltransferase activity of the vaccinia virus capping enzyme D1 subunit is stimulated by the D12 subunit. Identification of amino acid residues in the D1 protein required for subunit association and methyl group transfer.' Mao X, Shuman S. *J Biol Chem.* 1994 Sep 30;269(39):24472-9.

In sum, many of the references are wrong and the whole manuscript needs to be carefully scrutinized to make sure all citations in the text are accurate with respect to the numbered reference list.

Once these points are resolved, the revised paper should be acceptable for publication in *Nat Comm*

All done!